# Brain microvasculature defects and Glut1 deficiency syndrome averted by early repletion of the glucose transporter-1 protein

Maoxue Tang[1,2], Guangping Gao[3,4], Carlos B. Rueda[2,5,6], Hang Yu[7], David N. Thibodeaux[7], Tomoyuki Awano[1,2], Kristin M. Engelstad[5,6], Maria-Jose Sanchez-Quintero[6], Hong Yang[5,6], Fanghua Li[5,6], Huapeng Li[3,4], Qin Su[3,4], Kara E. Shetler[5,6], Lynne Jones[8], Ryan Seo[9], Jonathan McConathy[10], Elizabeth M. Hillman[7], Jeffrey L. Noebels[9], Darryl C. De Vivo[2,5,6] & Umrao R. Monani[1,2,6]

Haploinsufficiency of the *SLC2A1* gene and paucity of its translated product, the glucose transporter-1 (Glut1) protein, disrupt brain function and cause the neurodevelopmental disorder, Glut1 deficiency syndrome (Glut1 DS). There is little to suggest how reduced Glut1 causes cognitive dysfunction and no optimal treatment for Glut1 DS. We used model mice to demonstrate that low Glut1 protein arrests cerebral angiogenesis, resulting in a profound diminution of the brain microvasculature without compromising the blood–brain barrier. Studies to define the temporal requirements for Glut1 reveal that pre-symptomatic, AAV9-mediated repletion of the protein averts brain microvasculature defects and prevents disease, whereas augmenting the protein late, during adulthood, is devoid of benefit. Still, treatment following symptom onset can be effective; Glut1 repletion in early-symptomatic mutants that have experienced sustained periods of low brain glucose nevertheless restores the cerebral microvasculature and ameliorates disease. Timely Glut1 repletion may thus constitute an effective treatment for Glut1 DS.

[1] Department of Pathology & Cell Biology, Columbia University Medical Center, New York, New York 10032, USA. [2] Center for Motor Neuron Biology and Disease, Columbia University Medical Center, New York, New York 10032, USA. [3] Department of Microbiology and Physiological Systems, University of Massachusetts Medical School, Worcester, Massachusetts 010605, USA. [4] Horae Gene Therapy Center, University of Massachusetts Medical School, Worcester, Massachusetts 010605, USA. [5] Colleen Giblin Laboratory, Columbia University Medical Center, New York, New York 10032, USA. [6] Department of Neurology, Columbia University Medical Center, New York, New York 10032, USA. [7] Laboratory for Functional Optical Imaging, Departments of Biomedical Engineering and Radiology, Mortimer B. Zuckerman Mind Brain Behavior Institute and Kavli Institute for Brain Science, Columbia University, New York, New York 10027, USA. [8] Department of Radiology, Washington University School of Medicine, St. Louis, Missouri 63110, USA. [9] Department of Neurology, Baylor College of Medicine, Houston, Texas 77030, USA. [10] Division of Molecular Imaging and Therapeutics, University of Alabama, Birmingham, Alabama 35249, USA. Correspondence and requests for materials should be addressed to D.C.D. (email: dcd1@columbia.edu) or to U.R.M. (email: um2105@columbia.edu).

Mutations in the *SLC2A1* gene evolve into the rare but often incapacitating pediatric neurodevelopmental disorder, Glut1 deficiency syndrome (Glut1 DS)[1,2]. Initially considered exceptionally rare, reports that *SLC2A1* mutations account for ~1% of idiopathic generalized epilepsies and the recognition of an expanding Glut1 DS phenotype suggest that there may be in excess of 11,000 individuals afflicted with the disorder in the US alone[3,4]. Patients with classic Glut1 DS suffer low brain glucose levels and exhibit a phenotype characterized by early-onset seizures, delayed development, acquired microcephaly (decelerating head growth) and a complex movement disorder combining features of spasticity, ataxia and dystonia[5,6]. Low concentration of glucose in the cerebrospinal fluid (CSF), also known as hypoglycorrhachia, is the most reliable biomarker of the disease[2]. The disease characteristics of Glut1 DS are consistent with the widespread but especially abundant expression of Glut1 in the endothelial cells (ECs) of the brain microvasculature[7], where the protein facilitates the transport of blood glucose across the blood–brain barrier (BBB) to the CNS.

Although the genetic cause of Glut1 DS was identified almost two decades ago and notwithstanding widespread interest in Glut1 biology, little is known about the precise molecular and cellular pathology underlying the human disorder. Nor is there an optimal treatment for Glut1 DS. Clinicians have so far relied mostly on the ketogenic diet[8,9]. However, the diet is, at best, partially effective, mitigating seizure activity in some young patients but unable to attenuate virtually any other neurological deficit[10].

We modelled Glut1 DS in mice by inactivating one copy of the murine *Slc2a1* gene[11]. Mutants display many of the signature features of human Glut1 DS including seizure activity, hypoglycorrhachia, micrencephaly and impaired motor performance. Here we link overt manifestations of brain dysfunction in the mutants to profound defects of the cerebral microvasculature. We demonstrate that low Glut1 protein not only delays brain angiogenesis but also triggers microvasculature diminution, without affecting BBB integrity. Repletion of the protein in neonatal Glut1 DS model mice ensures the proper development of the brain microvasculature and preserves it during adulthood. Moreover, seizures and disease progression in these mice is rapidly arrested. Restoring the protein to 2-week old mutants, in which certain disease characteristics are readily apparent, is less effective in shaping normal brain microvasculature. Yet, low brain glucose levels in the mice are reversed and an overall salutary effect of the intervention is observed. In contrast, initiating protein repletion in symptomatic, adult (8-week old) mice raises brain glucose levels but fails to either mitigate brain microvasculature defects or attenuate major Glut1 DS disease features. We conclude that adequate Glut1 protein is indispensable for the proper development and maintenance of the capillary network of the brain. We further conclude that there is a limited postnatal window of opportunity to treat Glut1 DS using Glut1 augmentation as a therapeutic strategy. Nevertheless, timely reinstatement of the protein proves highly effective in preventing, indeed reversing, aspects of the disorder in the symptomatic individual.

## Results

**Brain microvasculature defects in Glut1 DS model mice.** Brain dysfunction is a characteristic feature of Glut1 DS patients and model mice. Moreover, the Glut1 protein is abundantly expressed in endothelia lining the brain microvasculature. We therefore began our investigation by examining the cerebral capillary network of mutant and control mice using fluorescently labelled lectin and an antibody against Glut1. As the structures identified by the two probes were invariably in perfect register (Supplementary Fig. 1a), further quantification of the microvasculature relied on lectin staining alone. This was carried out on 2-week, 8-week and 20-week old mice. We began by examining the capillaries in the thalamus, as this region appears particularly hypometabolic in positron emission tomography (PET) scans[12–14]. We found that the density of the capillaries, as determined by the total length traversed by them within a defined volume, was no different in mutants and WT mice at 2 weeks of age (Fig. 1a,b). Nor were there differences in the distribution of the sizes of individual capillaries or vessel branch points between the two cohorts of mice (Supplementary Fig. 1b,c). Brain angiogenesis continued in WT mice over the following eighteen weeks so that the capillary network had expanded by ~27% in the thalami of 8-week animals and a further ~5% by 20 weeks of age. In striking contrast, the capillary network of 2-week old Glut1 DS mice not only failed to expand, but rather diminished in size over the next eighteen weeks, appearing decidedly fragmented in the end. The microvasculature network in 8-week and 20-week old mutants was thus ~30% and ~40% respectively smaller than in age-matched controls (Fig. 1b). This diminution was not merely a consequence of a decrease in the overall extent of the capillary network, but additionally derived from smaller individual lectin-stained vessels with fewer branches (Fig. 1c,d). To ascertain if the diminished size of the capillary network in the thalami of mutants applied more generally to the cerebral microvasculature, we examined two additional regions—the cortex (primary motor and somatosensory cortex) and hippocampus (CA1, CA3 and DG regions)—of the brain. We found that the abundance of micro-vessels in these regions was similarly reduced in 20-week old mutants (capillary density: WT cortex = 1,625 ± 36, mutant cortex = 1,201 ± 32; WT hippocampus = 1,092 ± 44, mutant hippocampus = 810 ± 30, $P < 0.001$ in each instance, $t$ test, $N \geq 3$ mice of each genotype). Overall, the results suggest that Glut1 is required both for the elaboration as well as the maintenance of the cerebral microvasculature.

Diminution of the brain microvasculature could compromise the integrity of the BBB and lead to extravasation of serum proteins[15,16]. To investigate this possibility, we first quantified serum and CSF concentrations of albumin and IgG in 5–6 month old Glut1 DS mice and WT littermates. An increase in the CSF to serum albumin ratio is suggestive of increased BBB permeability whereas a rise in the CSF index (ratio of CSF to serum IgG divided by the albumin ratio in these two compartments – to correct for variances in BBB permeability) is indicative of enhanced IgG synthesis in the CNS and possible inflammation or infection[17,18]. We found no increases in albumin ratio, IgG ratio, or CSF index in the Glut1 DS mice (Fig. 2a,b; Supplementary Fig. 2a). This argues against either a disruption of the BBB or CNS inflammation in adult Glut1 DS mutants.

Although there was no significant increase in either albumin or IgG ratio in the mutants, the latter parameter trended higher. We therefore applied a second method to re-examine BBB integrity in the mutants. Accordingly, mice were intravenously administered either fluorescently labelled albumin or IgG, and brain sections examined 16 h later for extravasation of labelled protein into the parenchyma. In neither experiment was fluorescence detected in brain parenchyma of mutant mice (Fig. 2c; Supplementary Fig. 2b). In contrast, and as expected, pre-treating animals with kainic acid, an excitotoxic agent known to disrupt the BBB[19,20], resulted in copious fluorescent label outside the brain capillaries (Fig. 2c; Supplementary Fig. 2b). This suggested once again that Glut1 deficiency does not significantly compromise the functional integrity of the BBB.

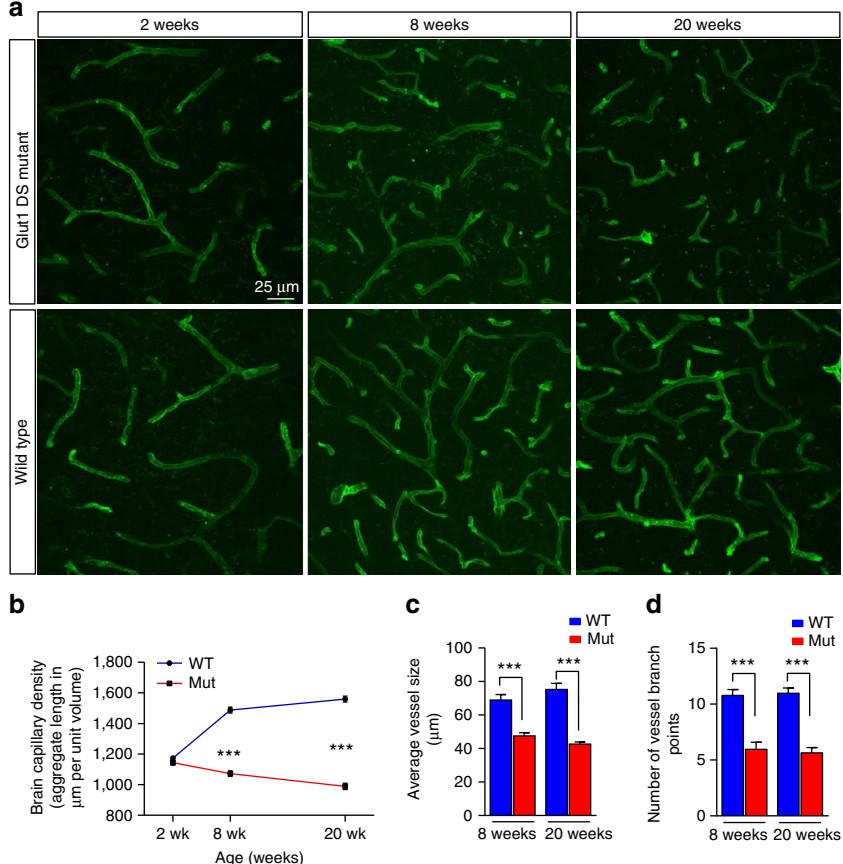

**Figure 1 | Cerebral microvasculature defects in Glut1 DS.** (**a**) Thalamic sections of Glut1 DS model mice and littermate controls stained with labelled lectin reveal cerebral angiogenesis defects and diminution of the microvasculature of mutants at 8 and 20 weeks of life. Graphs quantifying (**b**) aggregate cerebral capillary length, (**c**) mean vessel size and (**d**) average capillary branches in the mutants and controls illustrate the diminished size of the microvasculature in adult Glut1 DS mice. \*\*\*, $P < 0.001$, $t$-test, $n \geq 9$ regions from each of $N \geq 3$ mice of each genotype examined.

Recognizing that our experiments may not have detected subtle alterations in the Glut1 DS BBB, we carried out the following additional experiments. First, we substituted the albumin and IgG molecules with the labelled tracer, TMR-biocytin, which, owing to its much smaller size ($\sim 900$ daltons) is expected to traverse a leaky BBB with much greater ease, therefore enhancing the sensitivity of our assay[21]. Next, we used our model mice to examine expression levels of a subset of signature BBB endothelial genes, perturbations of which are known to disrupt the BBB[22–28]. Finally, we investigated CSF levels of proteins normally restricted to the serum in a cohort of Glut1 DS patients. Consistent with our earlier findings, we found no difference in levels of fluorescently labelled biocytin in the brain parenchyma of mutant and control animals, whereas abundant and intense signal was detected in the CNS of mice treated with the BBB-disrupting agent kainic acid (Supplementary Fig. 3a). In support of an intact BBB, as assessed by our labelled tracer assays, we detected no major alterations in the cerebral expression of Cldn3, Cldn5 and Cldn12 (tight junction proteins), Abcg2 and Abcb1b (active efflux transporters), Cdh5 (adherens junction protein) and Pvlap (plasmalemma vesicle-associated protein) (Supplementary Fig. 3b–h). Finally, we found no evidence of elevated CSF serum proteins and thus a compromised BBB in Glut1 DS patients evaluated by us between the ages of 6 months and 10 years (Mean CSF glucose in mg dl$^{-1}$: $32.58 \pm 0.57$, $N = 44$; Mean CSF lactate in mmol l$^{-1}$: $0.93 \pm 0.03$, $N = 44$; Mean CSF protein concentration in mg dl$^{-1}$:

$20.52 \pm 1.10$ versus $22 \pm 5$ (ref. 29) $P = 0.19$, one sample $t$ test, $N = 44$ and $N = 599$ respectively). We conclude that although there is a distinct loss of the brain microvasculature in adult Glut1 DS mice, the BBB in these animals and likely in human patients remains largely intact.

Although our results clearly demonstrate that Glut1 deficiency results in a poorly developed brain microvasculature, how this evolves is unclear. Accordingly, in a concluding set of analyses, we sought to explore potential mechanisms linking Glut1 to the process of angiogenesis. Two results appear to establish such a link. First, we discovered a sharp decline in levels of Vegfr2 in 2-week old Glut1 DS mutants. Transcripts as well as protein were diminished in expression. Importantly, the relative paucity of the molecule was restricted to blood vessels, remaining unchanged in the capillary-depleted fraction (parenchyma) of the mutant brain (Fig. 3a–c). Vegfr2 is a major positive-signal transducer expressed in ECs as capillaries form and expand, establishing it as a potential mediator of the microvasculature defects in Glut1 DS[30,31]. We also investigated the effects of Glut1 deficiency on glycolytic flux, a modest inhibition of which is known to perturb blood vessel formation[32,33]. To do so, we used astrocytic cells from Glut1 DS mutant mice and WT controls to assess rates of extracellular acidification (ECAR) and oxygen consumption (OCR), measures of glycolytic flux and mitochondrial respiration respectively[34]. Both basal glycolytic flux and maximum respiratory capacity declined significantly in mutant cells (Fig. 3d; Supplementary Fig. 4a,b). This suggests that

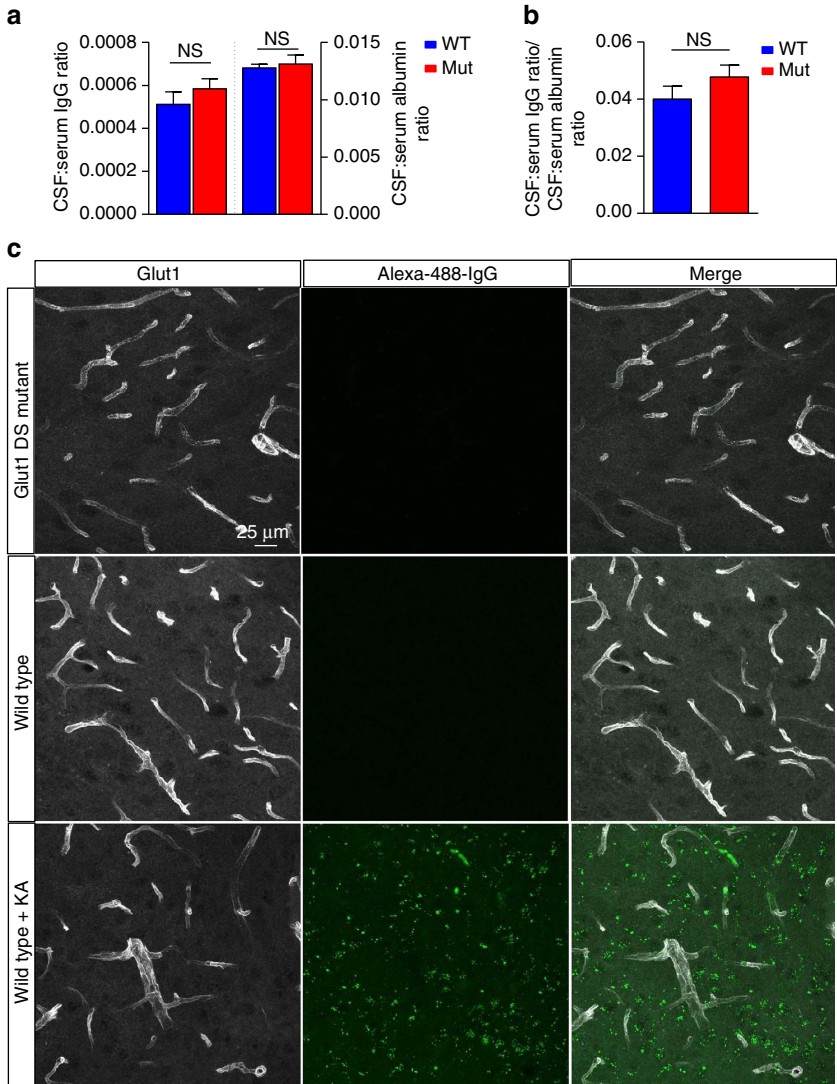

**Figure 2 | An intact blood–brain barrier in Glut1 DS model mice.** Evaluation of (**a**) CSF:Serum IgG and albumin ratios and (**b**) CSF index fails to provide evidence of any extravasation of serum proteins into the CSF of adult Glut1 DS mutant mice. $P > 0.05$, $t$-test, $N \geq 4$ mice of each genotype. (**c**) Labelled IgG systemically administered into Glut1 DS mutants does not escape into the neuropil. Note green fluorescence outside the microvasculature following chemically induced disruption of the BBB.

Glut1 haploinsufficiency does indeed impact glycolysis and lactate release, establishing a second potential link between Glut1 paucity and a diminished microvasculature.

**Glucose uptake restored in patient cells induced to express Glut1.** To determine if Glut1 repletion reverses or halts brain micro-vasculature defects and to explore the feasibility of this approach as a broader strategy to treat Glut1 DS, we sought to deliver and restore the protein to mutant mice. Adeno-associated virus 9 (AAV9) has emerged as an efficient vector to deliver therapeutic genes to target tissues[35–39]. Accordingly, we resolved to exploit AAV9 as a therapeutic vector. First, we investigated our constructs *in vitro* in fibroblasts from a Glut1 DS patient found to express low Glut1 protein. Murine (mGlut1) as well as human Glut1 (hGlut1) expression constructs were prepared by cloning the respective cDNAs into recombinant AAV plasmids harboring a chicken β-actin promoter element. Glut1 DS fibroblasts were then transfected with mGlut1, hGlut1 or an eGFP-containing construct. Each of the Glut1 constructs but not eGFP increased

Glut1 levels in the fibroblasts (Fig. 3e; Supplementary Fig. 5a). To determine if the construct-derived Glut1 was functional, we subjected fibroblasts co-transfected with one or the other Glut1 construct and an eGFP plasmid to a glucose uptake assay[40,41]. Uptake of 2-deoxyglucose (2-DOG), a labelled glucose analog, was indeed enhanced by the Glut1 constructs, approaching levels in control cells (Fig. 3f; Supplementary Fig. 5b). This suggested that the proteins produced from the constructs were functional.

**Phenotypic rescue following early Glut1 repletion.** Having demonstrated the functionality of the Glut1 constructs *in vitro*, we proceeded to test the effect of restoring protein in model mice. For this, we selected the mGlut1 construct. As a prelude to this and our later experiments, we examined the distribution of an AAV9-eGFP construct following systemic administration of the virus into postnatal day 3 (PND3) Glut1 DS mice. As expected, eGFP fluorescence was found in all major organ systems at 1-month. Importantly, this included the brain

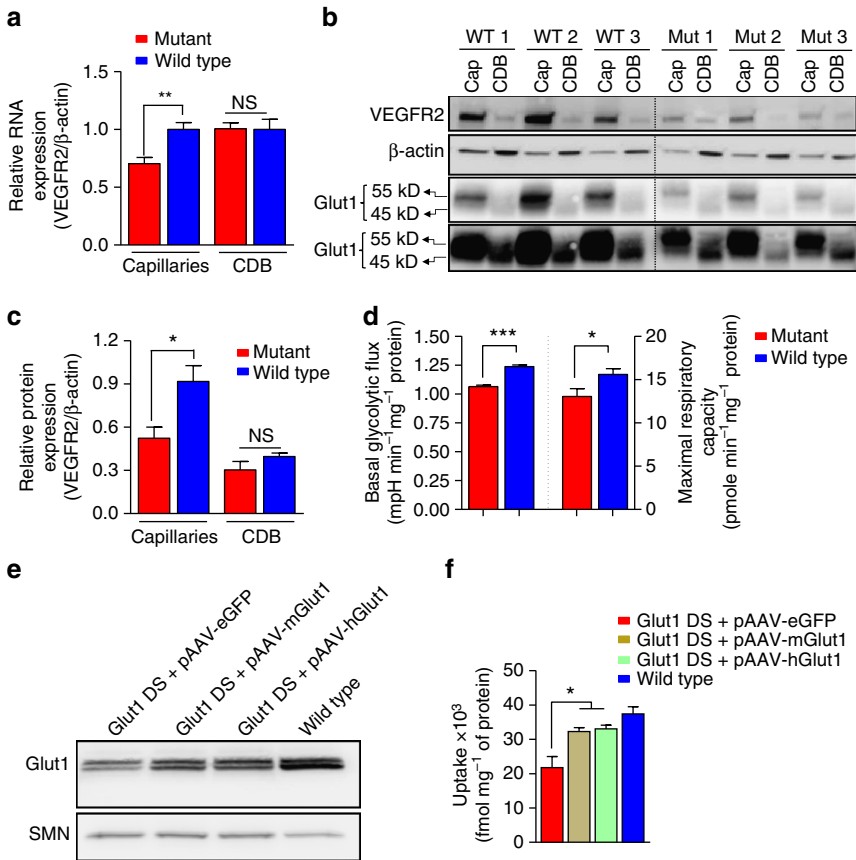

**Figure 3 | Potential mediators of angiogenesis defects in Glut1 DS and a functional evaluation of Glut1 constructs for gene replacement.**
(**a**) A quantification of Vegfr2 RNA levels demonstrates reduced transcripts specifically in the capillaries of the Glut1 DS brain. (**b**) Western blot analysis and (**c**) a quantification of Vegfr2 protein in the capillaries (Cap) and the capillary-depleted brain (CDB) fraction also depict reduced protein in mutant brain vessels. Also shown in the blot are corresponding Glut1 levels which are particularly high in the capillaries (55 kDa isoform) compared with the CDB fraction (45 kDa isoform). Results of high as well as low exposure times are depicted. (**d**) Basal glycolytic flux and maximal respiration are both significantly compromised in cells from Glut1 DS model mice. Panels a–d: *, **, $P < 0.05$, $P < 0.01$, t-test, $N \geq 3$ samples, 2 independent preparations. (**e**) Western blot depicting an increase in Glut1 protein following transfection of Glut1 DS patient fibroblasts with either a murine (lane 2) or human (lane 3) Glut1 cDNA construct. The ubiquitously expressed SMN protein was used as a loading control. (**f**) Quantitative representation of the levels of a radio-labelled glucose analogue, 2-DOG, taken up by cells transfected with the Glut1 constructs. Note the increase in the labelled 2-DOG in cells that were transfected with the Glut1 constructs. Also note, that pAAV denotes the fact that the relevant construct was a plasmid. *, $P < 0.05$, $N \geq 3$ assays, one-way ANOVA.

microvasculature (Supplementary Fig. 6a). Accordingly, we next administered a cohort of PND3 Glut1 DS mice either $4.2 \times 10^{11}$ GC of AAV9-Glut1 or vehicle alone. Wild-type mice administered vehicle alone served as controls. Six to eight weeks later, motor performance was examined. As expected from previous work[11], mutants treated with vehicle alone performed very poorly on a rotarod relative to WT littermates. In contrast, AAV9-Glut1-treated mutants exhibited a dramatic improvement—at all time points tested (Fig. 4a). In a second assay for motor performance, the vertical pole test, the Glut1-treated mutants performed as proficiently as WT mice (Fig. 4b). Mutants administered AAV9-Glut1 through a different route—into the intracerebral ventricles—also displayed improved motor performance, whereas no additional benefit accrued from over-expressing Glut1 in WT mice (Supplementary Fig. 6b). Unless otherwise noted, subsequent results stem from systemically administered virus.

To determine if the improved motor performance of the AAV9-Glut1-treated mutants was a consequence of increased Glut1 expression, we quantified the transcript as well as the protein in brain tissue of 20-week old mice. This time point also

marked a period, in mutants as well as wild-type animals, during which Glut1 mRNA levels stabilized (Supplementary Fig. 6c). QPCR experiments demonstrated that Glut1 transcripts were indeed increased in the treated mice, either approaching or, in some instances, exceeding levels in WT littermates (Fig. 4c). Expression of Glut1 also increased in other tissues of AAV9-Glut1-treated mice (Supplementary Fig. 6d–g). Western blot analysis of Glut1 protein in brain tissue reflected the results of the QPCR experiments (Fig. 4d,e), and further demonstrated that the repletion experiments resulted in concomitant increases in both astrocytic (45 kDa) as well as endothelial (55 kDa) isoforms of the Glut1 protein. These results provide a molecular basis to the mitigation of the motor defects observed in treated model mice.

To determine whether early repletion of Glut1 had rescued or prevented hypoglycorrhachia and micrencephaly, CSF glucose and brain weight respectively were examined at 20-weeks in virus-treated mice and relevant controls. As a prelude to these experiments and to obtain baseline values for subsequent analyses, we measured blood and CSF glucose levels as well as brain and body weights periodically between 1 week and 20 weeks

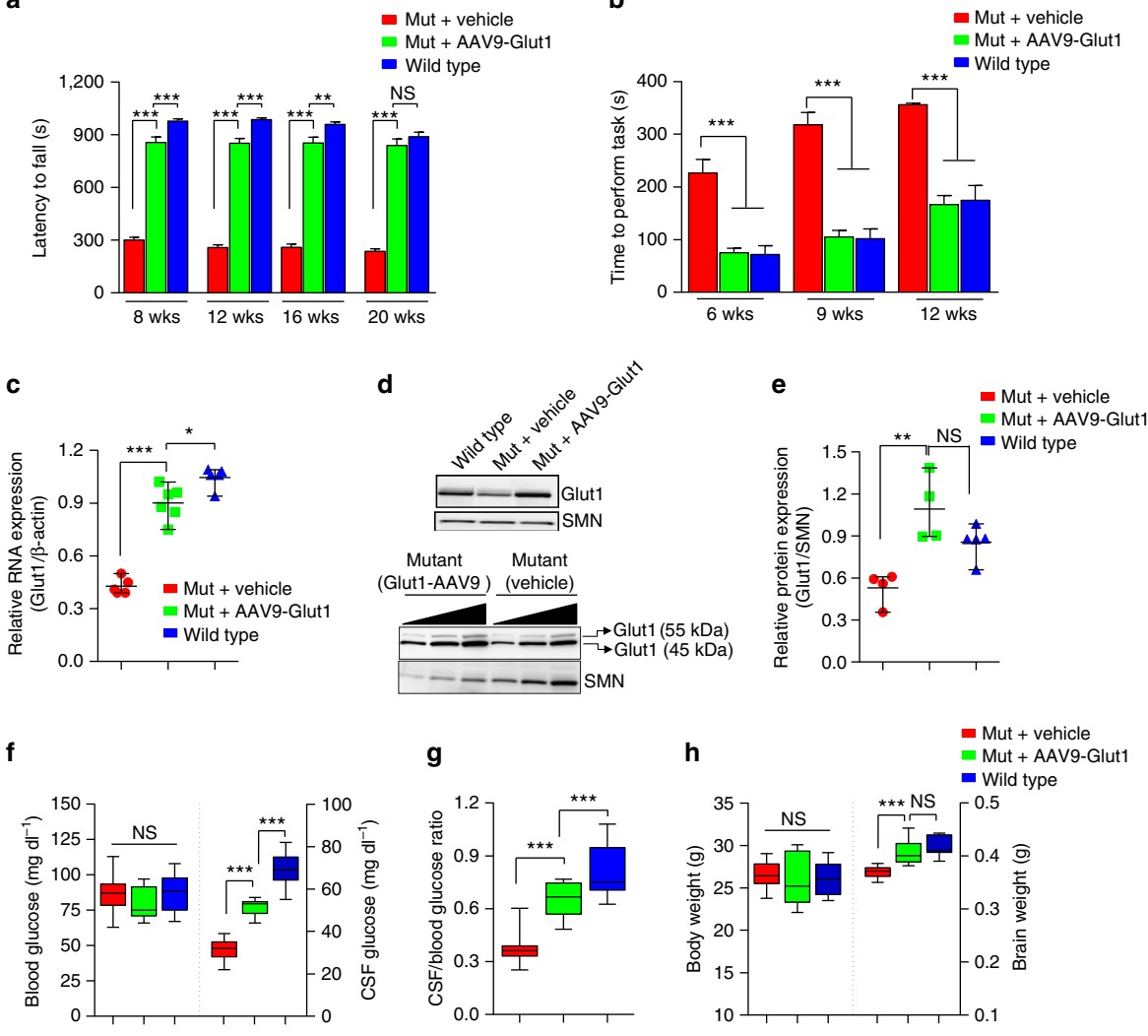

**Figure 4 | Early Glut1 repletion mitigates the Glut1 DS disease phenotype in model mice.** (**a**,**b**) Systemically administered AAV9-Glut1 results in a significant improvement in motor performance on (a) the rotarod and (b) in a vertical pole assay. $N \geq 16$ mice in each cohort. (**c**) Quantification of RNA (endothelial and astrocytic) expression in brain tissue of controls and mutants treated on PND3. $N \geq 5$ mice in each cohort. (**d**) Representative western blot analysis of Glut1 protein in brain tissue of Glut1 DS mutants following repletion of the protein. The lower set of blots demonstrates that repletion of Glut1 by the virus raises the 45 kDa (astrocytic) as well as 55 kDa (endothelial) differentially glycosylated isoforms of the protein. (**e**) Quantitative assessment of total protein (astrocytic and endothelial) in treated mutants and control mice. $N \geq 4$ mice each. (**f**,**g**) CSF, but not blood glucose levels are significantly increased in mutants treated with AAV9-Glut1; a corresponding increase in the ratio of CSF:Blood glucose was assessed. $N \geq 13$ mice in each cohort. (**h**) Microcephaly is ameliorated in mutants restored for Glut1 protein. $N \geq 7$ mice in each cohort. Note: *, **, ***, $P < 0.05$, $P < 0.01$, $P < 0.001$ respectively, one-way ANOVA for each panel in the figure.

in naive mutant and WT animals. Whereas blood glucose levels did not differ significantly in mutants at any of the time points chosen for evaluation, CSF glucose levels in the animals were dramatically reduced as early as 1 week of age (Supplementary Fig. 7a,b). Moreover, in contrast to CSF glucose concentrations in WT mice which peaked at ~3 weeks of age, remaining steady thereafter for the remainder of the evaluation period, levels in mutants declined significantly between 3 and 20 weeks of age (Supplementary Fig. 7b). This trend was reflected in CSF:blood glucose values (Supplementary Fig. 7c), consistent with a diminution of the brain microvasculature following the period of weaning and suggestive of an exacerbation of Glut1 DS with age. Overall body weight of mutant mice mirrored that of WT littermates over the period of assessment (Supplementary Fig. 7d). In contrast, a significant reduction in brain weight was detected in the Glut1 DS mice as early as 2 weeks of age

(Supplementary Fig. 7e). This difference persisted into adulthood. Collectively, the results reveal a lag between discernible hypoglycorrhachia and the appearance of micrencephaly, suggesting that Glut1 paucity and, consequently, reduced cerebral glucose conspire to retard the development of the brain relatively early in postnatal life.

Having determined the evolution of hypoglycorrhachia and micrencephaly in untreated mutants, we assessed these parameters in mutants administered AAV9-Glut1. CSF glucose levels in these mice had increased by ~64% relative to those in age-matched vehicle-treated mutants, while the CSF:blood glucose ratio, a more physiologically relevant parameter had risen even further by ~76%, approaching ~80% of WT values (Fig. 4f,g). Furthermore, and consistent with changes in brain weight lagging reductions in CSF glucose, micrencephaly in the treated mutants had either been prevented from developing

and/or completely reversed (Fig. 4h). A similar mitigation of these parameters was observed in mutants treated via the intracerebral ventricles (Supplementary Fig. 7f–h). Importantly, mice systemically administered AAV9-eGFP did not differ with respect to these two parameters from vehicle-treated animals, ruling out the possibility of a therapeutic effect of virus alone (Supplementary Fig. 7f–h). Together, these results attest to the marked therapeutic effect of early Glut1 repletion on disease.

**Early Glut1 repletion normalizes microvessels and reduces seizures.** Glut1 DS impairs brain glucose uptake[11–13]. Accordingly, we next ascertained if restoring Glut1 to mutant mice had reversed this impairment. Mutants treated with AAV9-Glut1 virus at PND3 and relevant controls underwent dynamic PET imaging at 12 weeks of age after the intravenous administration of 200–235 μCi of [18F] fluorodeoxyglucose (FDG), a glucose analogue widely used for clinical imaging. As expected, radioactive signal in brain tissue of anesthetized, vehicle-treated mutants was significantly lower than that in brains of WT mice, a finding congruent with decreased metabolism in the brain parenchyma of Glut1 DS mice (Fig. 5a). In contrast, signal in brain tissue of AAV9-Glut1 mice was enhanced and restored to levels seen in WT mice (Fig. 5a,b). This result is consistent with the outcome of our other experiments and suggests that early repletion of Glut1 protein prevents or may indeed rescue an impaired ability of the Glut1 DS organism to transport blood glucose into the cerebral parenchyma.

Finally, we asked if early Glut1 repletion had allowed for proper brain angiogenesis, had ensured the maintenance of the normal cerebral microvasculature and had prevented the onset of epileptic seizure-like activity characteristic of Glut1 DS. To address the first question we examined the capillary network in the brains (thalamus, cortex and hippocampus) of AAV9-Glut1-treated mice and controls at 20 weeks of age. Coronal sections stained with fluorescently labelled lectin showed that the architecture of the capillary network in the brains of AAV9-Glut1-treated mice was much more intricate than that of vehicle-treated mutants, indeed as elaborate as that of WT mice (Fig. 5c). When we quantified the capillaries in the different brain regions of the three groups of mice we were able to demonstrate that microvasculature density in the AAV9-Glut1-treated mice was indeed equivalent to that of WT mice and, expectedly, greater than that of their untreated cohorts (Fig. 5d). To confirm that the observations that we had made of the microvasculature in fixed tissue applied to normally perfused vessels as well, we resorted to an *in vivo* imaging technique that enabled us to assess the brain capillary network in live mice[42]. Following administration of labelled dextran into the tail veins of the mice, we exploited 2-photon microscopy to examine vessels within the somatosensory cortex of the three cohorts of animals. In accordance with earlier results, we found that whereas a significant diminution of the capillary network appeared in vehicle-treated Glut1 DS mutants, the size and complexity of the network was restored in mutants administered the AAV9-Glut1 vector (Fig. 5c,e,f; Supplementary Movies 1–3). We concluded our analysis of the effects of early Glut1 repletion by assessing brain activity in the three cohorts of mice. Glut1 deficiency triggers epileptic seizure-like activity as assessed by electro-encephalograms (EEGs)[11,14]. This defect was also mitigated by early Glut1 repletion. Thus, whereas vehicle-treated mutants continued to exhibit frequent seizures, Glut1-treated model mice were either seizure-free or experienced far fewer abnormal EEG events (Supplementary Fig. 7i,j).

Since these pre-clinical studies could serve as a springboard for the treatment of human Glut1 DS, we were interested in determining if Glut1 repletion triggered any deleterious effects in the major organ systems of the body. We furthermore investigated if expression of the virally delivered Glut1 transgene was sustained over time. A histochemical analysis of the major organ systems of the treated mutants stained with hematoxylin/eosin showed that except for subtle evidence of centrolobular steatosis in the liver, cellular morphology appeared grossly normal (Supplementary Fig. 8a). The steatosis likely derives from high expression of construct-derived Glut1 in the liver, transport of blood glucose into this organ and eventual conversion of the glucose to lipids. QPCR experiments to examine Glut1 expression longitudinally in neonatally treated mutants indicated sustained and robust expression of the transgene as late as 8 months of age (Supplementary Fig. 8b), suggestive of relatively low turnover of brain cells transduced by the virus and therefore a limited requirement for repeated administration of the therapeutic virus. In aggregate, our results indicate that Glut1 DS model mice, treated as neonates with AAV9-Glut1, recover and/or are prevented from becoming fully symptomatic. Early repletion of the Glut1 protein may therefore have a similar outcome in human Glut1 DS patients.

**A restricted therapeutic window in Glut1 DS model mice.** Considering the therapeutic effects of early Glut1 repletion and the need to treat the symptomatic individual, we resolved to use our model mice to attempt to define the temporal requirements for Glut1. Accordingly, we selected two additional time points to restore the protein. The first—2 weeks of age—was chosen as it is the start of the period when mice rapidly transition from a high-fat (∼30%; ketogenic) diet[43] derived exclusively from milk to a solid diet which comprises greater levels (>55%) of carbohydrates and relatively little (5–9%) fat[44]. Coincidentally, this marks the beginning of a significant decline in the expression of the *Mct1* gene[45,46], a concomitant increase in Glut1 expression[7] and initial evidence of retarded brain growth in our mutants. The second time point—8 weeks of age—marks a period during which Glut1 DS mutants are fully symptomatic. Mutants, at each of these time points were systemically administered AAV9-Glut1. The outcome was tested by first examining motor performance on the rotarod. To enable meaningful comparisons to mutants in which Glut1 repletion had been effected at PND3, testing was performed at identical time points. Interestingly, we found that mutants treated at 2 weeks of age performed significantly better than their vehicle-treated counterparts at all time points examined (Fig. 6a). In fact, between 8 weeks (PND3-treated: 857 s ± 31 s; 2-wk-treated: 868 s ± 35 s; $n \geq 15$, $P = 0.8$; *t*-test) and 12 weeks (PND3-treated: 851 s ± 28 s; 2-wk-treated: 792 s ± 46 s; $n \geq 15$, $P = 0.28$; *t*-test) of age, they performed as well as mutants treated at PND3. In marked contrast, restoring Glut1 at 8 weeks of age bestowed no significant benefit at any time point to the mutants (Fig. 6a). This result provides initial evidence of a limited therapeutic window of opportunity to restore Glut1 as a means of treating Glut1 DS.

To ensure that poor motor performance in mice treated at 8 weeks was not merely a consequence of poor transduction efficiency and thus poor expression caused by introducing the virus during adulthood, we assessed Glut1 mRNA and protein levels in brain tissue of the mice at 20 weeks of age. Expression in animals treated at 2 weeks was similarly examined. We found that Glut1 expression in the two cohorts of mutants was significantly greater than that in vehicle-treated mutants and at least as high as that in WT controls (Fig. 6b–d), suggesting that low transgene expression is unlikely to explain the poor outcome in mutants treated at 8 weeks.

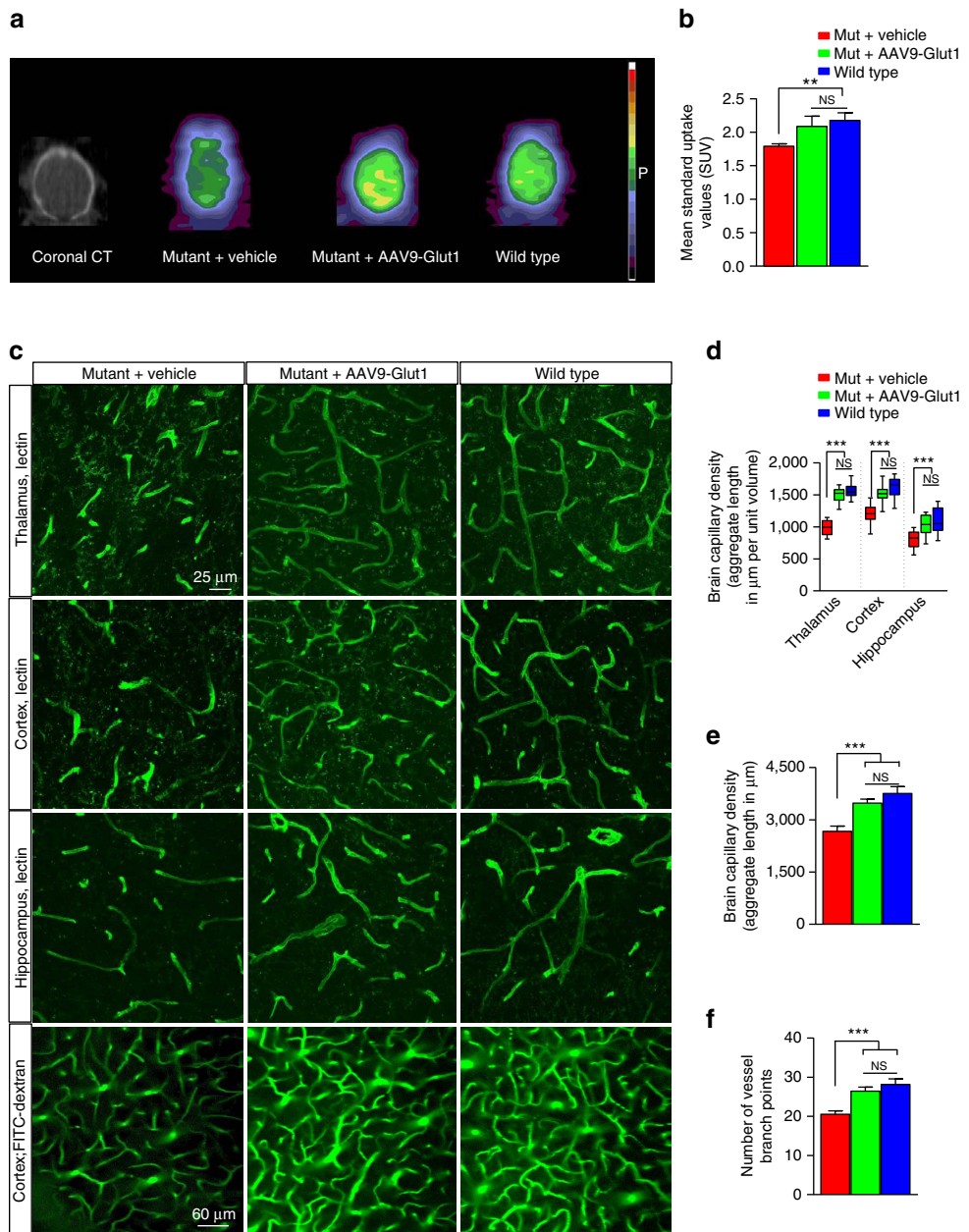

**Figure 5 | Normal cerebral microvasculature and brain glucose uptake in mutants treated early with AAV9-Glut1.** (**a**) Representative coronal images of small PET scans at 3–6 min following administration of $^{18}$F-FDG, show reduced uptake in the brain of a mutant mouse relative to WT and mutant mice treated with AAV9-Glut1. (**b**) Mean brain standardized uptake values (SUVs) at 3–6 min after injection of $^{18}$F-FDG in WT controls (SUV = 2.2 ± 0.11) and mutants treated with either vehicle (SUV = 1.8 ± 0.04) or AAV9-Glut1 (SUV = 2.1 ± 0.15). **, $P < 0.01$, one-way ANOVA, $N \geq 4$ mice in each cohort. (**c**) Immunohistochemistry or live-imaging experimental results of the brain microvasculature of model mice administered AAV9-Glut1 depicts a capillary network that is as elaborate and dense as that of WT, control littermates. Note reduced density and fragmented aspect of the brain capillaries in all three brain regions of vehicle-treated mutants. Also note (lower panels) the fewer FITC-Dextran perfused vessels in vehicle-treated but not AAV9-Glut1-treated model mice. Graphical representations of cerebral capillary densities of the three groups of mice following an analysis of (**d**) 4% PFA fixed tissue or (**e**,**f**) 2-photon live-imaging experiments. ***, $P < 0.001$, one-way ANOVA, $n \geq 9$ regions from each of $N \geq 3$ mice of each cohort.

To further analyse mutants administered virus at the two later time points, we assessed the effects of Glut1 repletion on CSF glucose levels and brain size at 20 weeks of age. We found that CSF glucose concentrations only increased appreciably in mice treated at 2 weeks of age (Fig. 6e). However, interestingly, CSF:blood glucose ratios increased in both cohorts of mice, a consequence of lowered blood glucose concentrations (Fig. 6f). The drop in blood glucose was also noted, albeit to a lesser extent, in mice treated at PND3 (Fig. 4f). Considering that the

measurements were made in fasting animals, one possible explanation of this outcome is that systemic expression of virus predisposes the animals, when fasted, to hypoglycemia—a likely result of high expression of the Glut1 transgene in organs such as liver and muscle where it contributes to elevated glucose uptake thus lowering serum glucose concentrations. Indeed, blood glucose concentrations in non-fasting mice treated with virus at PND3 or 2 weeks appeared no different from those in controls (Supplementary Fig. 9). Notwithstanding these

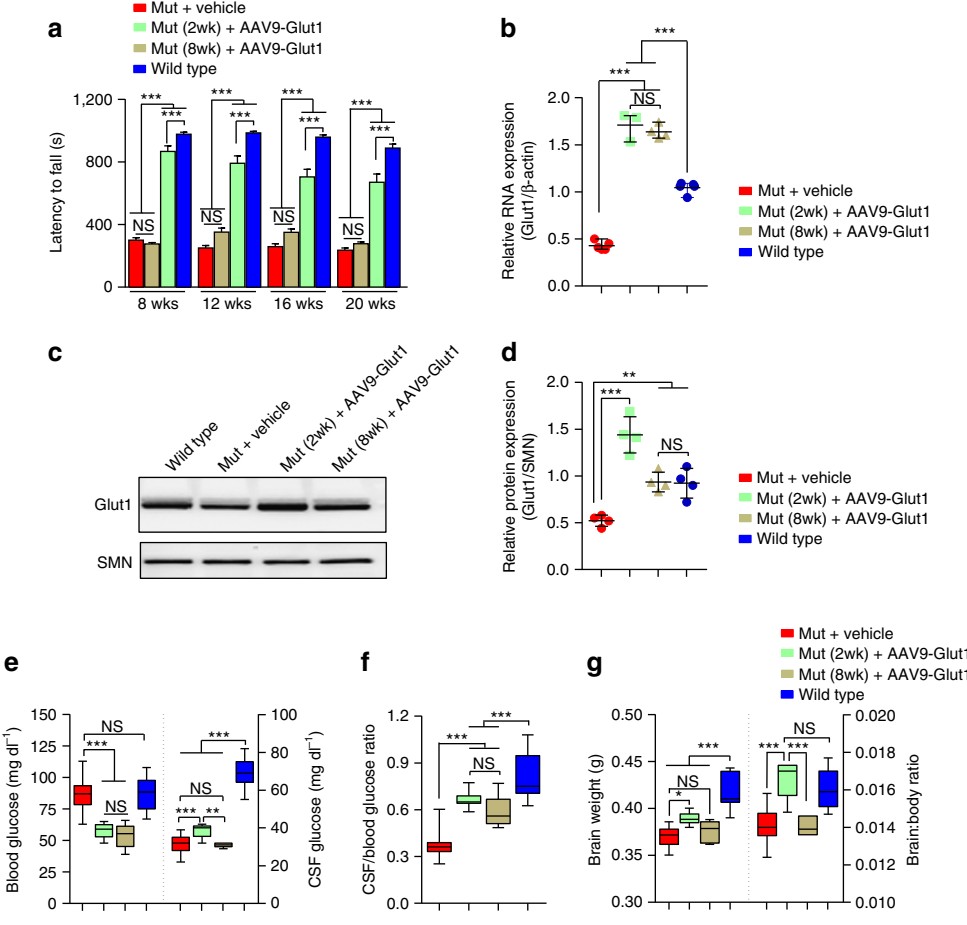

**Figure 6 | A defined window of opportunity to treat Glut1 DS.** (**a**) Rotarod tests reveal significantly improved motor performance of mutant mice treated with the therapeutic vector at 2 weeks but not 8 weeks of age. ***, $P < 0.001$, one-way ANOVA, $N \geq 8$ mice in each cohort. Late delivery of the AAV9-Glut1 to mutant mice does not preclude an increase in (**b**) Glut1 RNA expression or (**c,d**) Glut1 protein as assessed by western blot analysis. **, ***, $P < 0.01$ and $P < 0.001$, one-way ANOVA, $N \geq 4$ mice in each cohort. An evaluation of (**e,f**) blood and CSF glucose values and (**g**) micrencephaly in mutant mice treated at either 2 or 8 weeks of age. *, ***, $P < 0.05$ and $P < 0.001$, one-way ANOVA, $N \geq 8$ mice in each cohort.

findings, the aggregate results suggest that even late repletion of Glut1 protein is capable of raising CSF glucose levels relative to those in the blood.

When we examined the effects of restoring Glut1 on the micrencephalic Glut1 DS phenotype, we found that the brains of 2-week-treated mutants were significantly larger than those of vehicle-treated mice (Fig. 6g). This was even more evident when brain:body weight ratios were compared. In fact, corrected mean brain weight in mutants now appeared no different from that of WT mice (Fig. 6g). Given the obvious difference in this parameter between mutants and WT mice when virus was initially administered—at 2 weeks, the inability to detect it at 20 weeks suggests that the micrencephalic phenotype is reversible if Glut1 is restored in a timely manner. In contrast to the result in mice treated at two weeks, the micrencephaly persisted in the mice treated at 8 weeks. This suggests that even if relative CSF glucose concentrations are augmented by restoring Glut1 at this advanced stage of the disease, stimulating the expression of the protein fails to halt or reverse effects on brain volume.

**Earlier Glut1 repletion results in a more extensive capillary network.** To conclude our analysis of mutants treated at 2 and 8 weeks respectively, we assessed the effects of the treatment on the brain microvasculature at 20 weeks of age. Consistent with the persistence of an overt motor phenotype and reduced cerebral

volume in mice treated at 8 weeks, the microvasculature within the brains of the mutants appeared fragmented, less dense and greatly reduced in complexity relative to that of WT controls (Fig. 7a). In fact, qualitatively, it looked no different from that of age-matched, vehicle-treated mutants. Quantification of the density of the capillaries, the average size of the individually stained blood vessels and the frequency with which the capillaries branched respectively confirmed this to be the case (Fig. 7b–d). In contrast, and congruent with other parameters in the mutants, the brain microvasculature of mice treated at 2 weeks was significantly more elaborate than that of vehicle-treated or 8-week-treated mutants but less so than that of WT mice or mutants treated at PND3 (Fig. 7a,b). We noted similar differences when we quantified the average sizes of the capillaries and the number of occasions on which they branched except that the mean values obtained from mice treated at 2 weeks were statistically equivalent to those of PND3-treated and WT mice. Still, these results once again link Glut1 DS to brain microvasculature defects and demonstrate a marked correlation between the timing of Glut1 repletion, the ability to restore the cerebral capillary network and overall therapeutic effect realized by the mutant organism.

**Discussion**

Although the genetic cause of Glut1 DS was revealed almost two decades ago, relatively little progress has been made in identifying

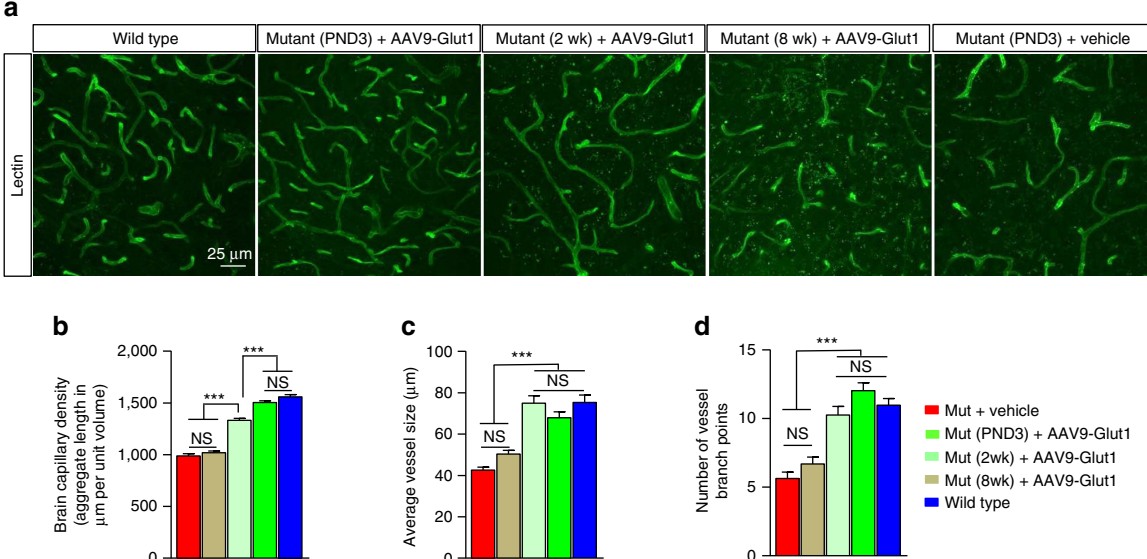

**Figure 7 | Brain capillary network restored following early but not late repletion of the Glut1 protein.** (**a**) Representative immuno-histochemical photomicrographs of the thalamic microvasculature in relevant controls and mutants treated with AAV9-Glut1 at various stages of the disease. Note persistent fragmentation and reduced complexity of the capillary network in vehicle-treated mutants and mutants treated at 8 weeks of age, an intermediate capillary density in cohorts treated at 2 weeks of age and a relatively normal microvasculature in mice administered virus at PND3. Quantitative estimation of (**b**) aggregate cerebral capillary length, (**c**) mean vessel length and (**d**) number of branches per capillary in the different cohorts of mice. ***, $P < 0.001$, one-way ANOVA, $n \geq 9$ regions from $N \geq 3$ mice in each cohort examined.

a truly effective treatment for the disorder, and much remains to be learned about the cellular pathology linking Glut1 mutations to the characteristic brain dysfunction seen in patients. Here, we attempted to address these deficiencies in model mice. Three principal findings emerge from our study. The first highlights novel defects of the brain microvasculature in the Glut1 deficient organism. These involve both a delay in the initial expansion of the cerebral capillary network as well as a later impairment in its maintenance. Thus in mature mutants, a striking diminution of the capillary network became evident. Remarkably, though, no discernible alteration in BBB integrity was noted. Our second salient finding demonstrates that gene replacement using an AAV9 vector constitutes a relatively straightforward, safe and highly effective means of treating Glut1 DS. Glut1 repletion in neonates had a major therapeutic effect on mutant mice, arresting the onset of a motor phenotype, enabling the development of the brain microvasculature and restoring parameters typically perturbed in the mutant to the wild-type state. Our final notable result addresses the temporal requirement for the Glut1 protein and allows one to define a window of opportunity to treat Glut1 DS by means of restoring the protein. Initiating a treatment in juvenile animals that were at least partially symptomatic, reversed hypoglycorrhachia, improved motor performance, accelerated the growth of the brain and facilitated the expansion of the cerebral microvasculature. Akin to effecting repletion during neonatal and early postnatal life, restoring the protein to the fully symptomatic adult mutant raised characteristically low brain glucose levels. However, in contrast to the outcome of early repletion, augmenting the Glut1 protein late failed to mitigate any other major feature of the Glut1 DS phenotype. We suggest that the accumulating damage sustained by the Glut1 deficient brain as it attempts to establish and refine important neural circuits eventually precludes the possibility of therapeutic rescue even if overall cerebral Glut1 expression is eventually restored. Our results predict that Glut1-deficient patients will be most responsive to gene replacement-type treatments relatively early

in the course of the disease. Nevertheless, there exists a period during the symptomatic phase of the disease when Glut1 DS can be effectively treated. These findings argue for the use of new-born screens to identify the pre-symptomatic Glut1 DS patient.

Although widely expressed, Glut1 is particularly abundant in the ECs of the brain microvasculature[47]. Moreover, brain dysfunction is a signature feature of Glut1 DS. Examining the capillary network of our mutant mice therefore appeared to be logical way to initiate our investigation. Still, we were startled by how profoundly loss of one copy of the *Slc2a1* gene had affected the elaboration of the brain microvasculature. Myriad genes govern CNS angiogenesis (ref. 48 and references therein), but few trigger such marked defects without a total ablation of their activities. Interestingly, the diminution of the brain capillary network that we observed became evident only in mature mice and did not compromise BBB integrity. The first finding—that of angiogenesis defects—is consistent with those of two prior reports, in which Glut1 deficient model fish and mice respectively were studied[49,50]. The second—defects of barriergenesis—is not. While the distinct findings in Glut1 deficient fish has a logical explanation and likely stems from a much greater (~90% versus ~50% in our study) level of Glut1 knockdown[49], the report of Winkler *et al.*[50] is more curious and difficult to reconcile with ours. In their study, massive (>10-fold) extravasation of serum proteins was observed as early as 2 weeks. Cerebral vasogenic edema ought to have followed but, surprisingly, was not reported. More importantly, such edema has neither been cited in the literature as characteristic of Glut1 DS patients nor detected by us over a three decade period in the clinic. These clinical observations correlate to a greater extent with our findings of an intact BBB in model mice than they do with those of Winkler and colleagues. Still, one oft-overlooked but important factor that might explain the distinct findings in the study of Winkler *et al.* is a difference in mouse strain background. Whereas our mice were maintained on a pure 129/SvJ background, the mutants in the study by Winkler

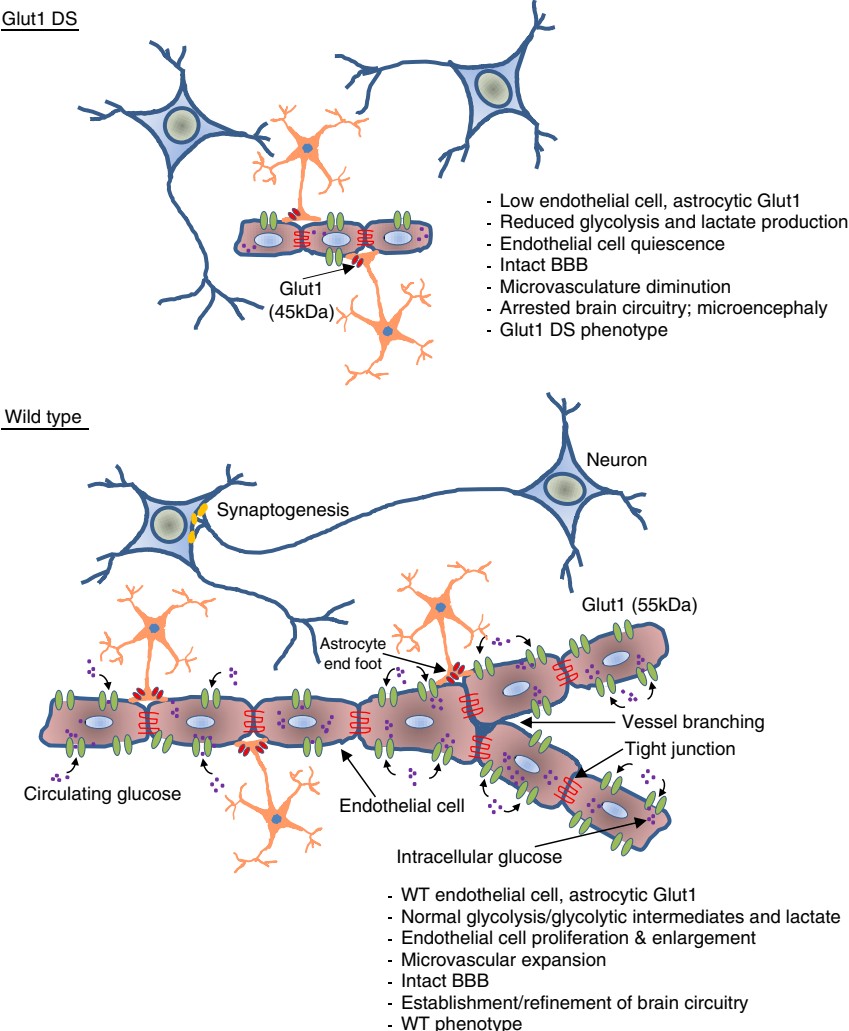

**Figure 8 | Relationship between brain microvasculature defects and Glut1 deficiency.** A model illustrating the direct as well as indirect effects of reduced Glut1 protein on brain microvascular ECs and the consequences of such a reduction first, on the capillary network and then on brain circuitry. Note that the BBB remains intact in the absence of overt cell loss and that ECs express 20–30 times more Glut1 than do astrocytes.

*et al.* derived from a mixed 129/SvJ-C57Bl/6J strain background. Genetic strain background is known to have profound effects on the disease phenotypes of model mice[51,52]. It is possible that strain-derived modifiers on the pure 129/SvJ background underlie the less severe BBB phenotype we report here. Testing this hypothesis will require additional studies.

The simultaneous existence of an intact BBB such as the one we report in our mutants and a simplified microvasculature appears puzzling at first, but is not necessarily surprising. Endothelial cells, particularly those at the leading edges of vessel sprouts (tip and stalk cells) so essential to the expansion of the microvasculature network, are known to be highly sensitive to metabolic or genetic perturbations[33,53]. For instance, perturbing Notch signalling or knocking down PFKFB3 profoundly affects the development of the microvasculature by potently inhibiting tip cell activity. Yet, ECs that constitute vessels that have already formed are relatively unaffected. Such phalanx cells merely enter a state of quiescence, exhibiting little evidence of ER, oxidative or energy stress and thus few signs of actual degeneration[33]. Accordingly, one might not expect the structural (BBB) integrity of the cells to be affected. Glut1 mutations could similarly perturb the activity of endothelial tip and stalk cells, becoming especially detrimental

to their ability to pioneer new vessels and expand the capillary network in response to the demands of a maturing brain. This chain of events involving both a proliferation ($\sim$PND0 to $\sim$PND15) as well as expansion ($\sim$PND10 to PND25) of ECs[54] is encapsulated, in part, in a model depicted in Fig. 8, although the precise insult(s) to neuronal circuitry remain(s) to be identified. It is to be noted that in addition to cell-autonomous perturbations within the ECs, it is likely that non-cell autonomous perturbations—originating in astrocytes and/or pericytes—impact the former, eventually hindering angiogenesis. One appealing, astrocyte-derived mediator is lactate, decreased levels of which have been shown to inhibit EC proliferation and vessel formation[55].

Our gene repletion studies and the identification of a therapeutic window of opportunity begin to define the temporal requirements for the Glut1 protein. Early restoration was clearly most effective but repletion at PND14 also had a marked effect. Subsequent augmentation, once the brain is considered mature, failed to provide benefit. These results are a logical outcome for a protein whose primary cellular site of action in ensuring proper brain development is assumed to be the cerebral microvasculature. These results are also a reflection of the normal postnatal development of the brain capillary network. In rodents, this

involves a proliferation of ECs that peaks at ~PND10 and subsides by ~PND15 (ref. 54). Subsequent increase in the size and complexity of the network, which eventually abates by 1 month, is mainly through the sprouting and elongation of existing cells. The expansion of the network coincides with an increase in expression of microvascular Glut1. Partial rescue of the Glut1 DS phenotype following gene replacement at PND14 is therefore expected and consistent with our findings. The maturation of the brain microvasculature is largely complete by ~1 month[54], likely reflecting the establishment of the adult complement of brain circuits. Interestingly, total cerebral Glut1 expression exhibits a decline during this period (Supplementary Fig. 7c). In this respect the requirements for Glut1 are not dissimilar to those of SMN, another widely expressed protein that is especially important in neuromuscular junction maturation and whose deficiency causes the motor neuron disease, spinal muscular atrophy[56,57]. Depleting SMN following neuromuscular junction maturation has a relatively benign effect[57]. Given the relative stability of the adult brain microvasculature, it would not be surprising if depleting Glut1 once the network is established is as innocuous.

This study has addressed important basic as well as clinically relevant aspects of Glut1 biology. However, as is often the case, it also raises interesting questions. For instance, what links reduced Glut1 to brain microvasculature defects? Plausible connections that have emerged from this study and will have to be pursued include reduced glycolytic flux and/or glycolytic intermediates, genes downstream of Glut1, for example, Vegfr2, or some combination thereof. Second, how do low Glut1 and a diminution of the cerebral capillary network lead to brain dysfunction? Micrencephaly is a well-established disease characteristic, but whether this is a consequence of overt cell loss, lower synaptic density or both, remains to be determined. In parallel, identifying specific regions of the brain that are affected will shed important light on the spatial requirements for cerebral Glut1. Finally, in spite of our observations of a grossly intact BBB in the Glut1 deficient organism, does protein haploinsufficiency trigger subtle, size-selective loosening of the barrier as reported in mice deficient in Cldn5? Notwithstanding these and other important questions raised here, our results serve as an important step in the quest to effectively treat Glut1 DS.

## Methods

**Primary cell culture experiments.** Before testing the efficacy of the Glut1 constructs in model mice, they were assessed in patient fibroblasts. To do so, $1 \times 10^6$ cells, derived from skin biopsies from patients or controls seen at the CUMC pediatric neurology clinic, were seeded in culture flasks, allowed to grow in M-106 medium (Life Technologies, Carlsbad, CA, USA) for ~48 h (~80% confluent) and then transferred to 6-well plates for transfection. Plasmid was delivered to the cells by electroporation using the Nucleofector kit (Lonza, Cologne, Germany). Astrocyte cultures for measurements of glycolytic flux were obtained from PND3 pups generated from crosses between Glut1$^{+/+}$ and Glut1$^{+/-}$ mice. Cerebral cortices were removed free of meninges, cut into small pieces, and enzymatically dissociated for 30 min in complete PBS medium without MgCl$_2$ and CaCl$_2$ (GIBCO-Invitrogen), 0.4 mg ml$^{-1}$ papain (Roche), and then mechanically dissociated, in the presence of DNase (Roche), by using glass pipettes of different pore size. Dissociated cells were collected by centrifugation (1,200g, 5 min). All cells used here were mycoplasma free.

**Glucose uptake assay.** Glucose uptake assays in the transfected cells and relevant controls were essentially carried out as reported by us in a prior study[41]. Briefly, ~48 h after transfection, the cells were incubated for 2 h in glucose-free DMEM, then washed twice in pre-warmed (37 °C) PBS and further incubated in the PBS for 15 min to achieve a zero-trans condition. Glucose uptake was initiated with a 500 μl solution containing a mix of $^{14}$C 2-DOG and unlabelled 2-DOG (Perkin Elmer, Waltham, MA, USA). Baseline glucose transport was assessed in parallel by subjecting one set of cells to 25 mM cytochalasin B (Sigma, St. Louis, MO, USA). The uptake reaction was terminated after a minute with stop solution (HgCl$_2$ + phloretin) (Sigma), the radioactive incubation medium discarded and the cells washed three times in rapid succession with 1 ml each of ice-cold stop solution.

Ice-cold ethanol (1 ml, 100%) was then added to the culture plates and the cells lysed in the solution for 30 min. The entire lysate and 5 mls of the Hionic Fluor cocktail (Perkin Elmer, Waltham, MA, USA) was then placed in a scintillation vial and $^{14}$C 2-DOG influx quantified in a scintillation counter (Tri-carb 2300 TR counter, Packard Biosciences, Meridien, CT). To correct for possible variations in cell number, cell pellets from the above experiment were digested (0.1N NaOH, 0.1% Triton X-100) and protein concentrations in 50 μl aliquots assessed using the Bio-Rad protein assay kit (Bio-Rad Labs, Hercules, CA, USA). Glucose uptake was expressed relative to total cell protein.

**Phenotypic evaluations and CSF/blood glucose measurements.** Glut1 DS mice were initially created at Columbia University and maintained on a 129/SvJ genetic background. Mutants for this project were generated by breeding mutant male mice with WT females, and identified by PCR as previously described[11]. For the behavioural studies GraphPad Prism was used to determine sample sizes to detect differences of at least two standard deviations with a power of 80% ($P < 0.05$). Mice were not randomized, but as mutants do not exhibit an overt disease phenotype, it was possible to blind the investigator to the particular cohort being assessed. Motor performance in the different cohorts of mice was evaluated by rotarod analysis and/or a vertical pole test. To administer the rotarod test, mice were subjected to a training period of 5 min on an accelerating rotarod (Ugo Basile Inc., Italy) three times a day for four consecutive days. Measurements were recorded on the fifth day at a setting of 25 r.p.m. Duration of time on the rotating rod was recorded and the experiment terminated if a mouse surpassed 1,000 s. The vertical pole test employed a protocol initially described by Matsuura et al.[58] and refined by us in a study to examine a mouse model of spinal muscular atrophy[57]. Briefly, mice were placed in the centre of a vertically positioned metal pole (60 cm long, 1 cm diameter) covered with a mesh tape, with their snouts oriented toward the ceiling. Motor performance was measured by the latency to turn around to descend the pole. To assess brain and body size, and investigate CSF/serum glucose levels, mice were first subjected to an overnight fasting period. Subsequently they were weighed, blood collected from the tail vein, CSF extracted from the cisterna magna as detailed by us in a prior study[11] and the brain removed and weighed. Glucose concentrations in the CSF and blood were assessed on disposable strips using a Contour Next EZ glucose meter (Bayer Corp., NJ, USA) and reflect levels in freely moving animals. Except for the determination of brain and body weights where results presented are from male mice alone, experiments described in this study included animals of either gender. To preclude strain background effects, littermate controls were used in all experiments. Additionally, all experiments were conducted in accordance with the protocols described in the NIH *Guide for the Care and Use of Animals* and were approved by the Columbia University Institutional Laboratory Animal Care and Use Committees.

**Virus production and administration.** Recombinant virus was produced as described below (also see ref. 59). Murine or human Glut1 cDNAs were initially inserted into a rAAV plasmid consisting of the vector genome, a CMV-enhanced chicken-β-actin promoter and AAV2 ITRs flanking the expression cassette. To generate virus, the plasmid carrying the mGlut1 cassette was co-transfected into HEK 293 cells with a packaging plasmid and an adenovirus helper plasmid. The packaging plasmid expresses AAV2 regulatory proteins and AAV9 capsid proteins. The regulatory proteins excise the recombinant genome from the rAAV vector plasmid, replicate the genome, and package it into AAV virions. Adenovirus serotype 5 E1, E2a and E4 proteins, and VA I and II RNAs expressed from the adenovirus helper plasmid provide helper functions essential for rAAV rescue, replication, and packaging. The packaged, recombinant viral particles were then purified by a CsCl gradient sedimentation method, desalted by dialysis and subjected to a quality control analysis as detailed elsewhere[59].

Delivery of the AAV9-Glut1 vector ($4.2 \times 10^{11}$ GC in 35 μl) into PND3 mice was accomplished through the retro-orbital sinus as reported[60]. Two week old mice were similarly administered virus ($1.2 \times 10^{12}$ GC in 100 μl) after rendering them unconscious with isoflurane. In contrast, adult mice were administered virus ($4 \times 10^{12}$ GC in 400 μl) through the tail vein. To do so, the mice were first restrained, the tail sterilized with alcohol and the tissue warmed under a brooder lamp for 5 min to dilate the tail vein and facilitate delivery of the virus. To deliver virus ($3 \times 10^{11}$ GC) into the intra-cerebral ventricles of mice we followed a protocol described by Glascock and colleagues[61].

**Quantitative PCR.** Glut1 and BBB signature gene transcript levels were determined by QPCR. RNA was extracted from the relevant tissue using Trizol (Life Technologies, Carlsbad, CA, USA) according to the manufacturer's instructions and 2 μg reverse transcribed following treatment with DNAse. QPCRs were carried out on a Realplex4 Mastercycler (Eppendorf, Germany) using Maxima SYBR green (Thermofisher Scientific, Waltham, MA, USA). All samples were run in triplicate and relative concentrations calculated using the ddCT method and by normalizing to β-actin (for Glut1 expression) or to GAPDH (for the BBB signature genes). PCR primers are indicated in the table (Table 1) below.

**Western blotting.** Glut1 protein levels in the various cohorts of mice were assessed by western blot analysis using standard techniques described

**Table 1 | PCR primers utilized to examine levels of BBB signature genes or transcripts altered in Glut1 DS.**

| Gene | Primer sequence |
|---|---|
| Glut1 F | 5′ CTT GCT TGT AGA GTG ACG ATC 3′ |
| Glut1 R | 5′ CAG TGA TCC GAG CAC TGC TC 3′ |
| Cldn3 F | 5′ GAT GGG AGC TGG GTT GTA CG 3′ |
| Cldn3 R | 5′ GAG GAT CTT GGT GGG TGC AT 3′ |
| Cldn5 F | 5′ GTT AAG GCA CGG GTA GCA CT 3′ |
| Cldn5 R | 5′ TAC TTC TGT GAC ACC GGC AC 3′ |
| Cldn12 F | 5′ GTC CTG TCC TTC CTG TGT GG 3′ |
| Cldn12 R | 5′ TGA ATG TGA TCA GCC GCA GT 3′ |
| Cdh5 F | 5′ ATT GGC CTG TGT TTT CGC AC 3′ |
| Cdh5 R | 5′ CAC AGT GGG GTC ATC TGC AT 3′ |
| Abcb1b F | 5′ GG GAG ATC CTC ACC AAG CG 3′ |
| Abcb1b R | 5′ CCC ATC GCC CCT TA ACA CT 3′ |
| Abcg2 F | 5′ TGG ACT CAA GCA CAG CGA AT 3′ |
| Abcg2 R | 5′ CGG AAG CCA GTA AGG TGA GG 3′ |
| Plvap F | 5′ CGT CAA GGC CAA GTC GCT 3′ |
| Plvap R | 5′ CAG CAG GGT TGA CTA CAG GG 3′ |
| β-actin F | 5′ TGT TAC CAA CTG GGA CGA CA 3′ |
| β-actin R | 5′ GGG GTG TTG AAG TCT CAA AA 3′ |
| Gapdh F | 5′ CGA CTT CAA CAG CAA CTC CCA CTC TTC C 3′ |
| Gapdh R | 5′ TGG TGG TCC AGG GTT TCT TAC TCC TT 3′ |
| Vegfr2 F | 5′ CTA GCT GTC GCT CTG TGG TT 3′ |
| Vegfr2 R | 5′ CTG TCC CCT GCA AGT AAT CTG A 3′ |

elsewhere[11,62]. Briefly, cells or tissues were lysed in lysis buffer containing protease inhibitors (Complete Protease Inhibitor Cocktail tablets and pepstatin each from Roche, Indianapolis, IA, USA) and 50 μg of total protein resolved by gradient SDS-PAGE (Bio-Rad Labs, Hercules, CA, USA). Glut1 rabbit polyclonal (1:5,000; Millipore; #CBL242) and SMN mouse monoclonal (1:5000; BD Biosciences; #610640) antibodies were probed respectively with HRP-linked donkey anti-rabbit (Santa Cruz Biotechnology; #sc2317) and sheep anti-mouse (GE Healthcare) secondary antibodies each diluted 1:5000 and visualized on an ImageQuant LAS 4000 machine (GE Healthcare; #NA-931) using the ECL Detection Kit (RPN 2109; GE Healthcare). Band intensities were determined using ImageJ software (NIH). Western blots of Vegfr2 were carried out using a rabbit monoclonal antibody (1:2,000; Thermofisher; #B.309.4) and a mouse monoclonal antibody against actin (1:5000; Sigma; #A5441). Note: western blot images have been cropped for presentation. Full-sized membrane strips probed with the respective antibodies are presented in Supplementary Fig. 10.

**Blood chemistry.** Mice were euthanized with $CO_2$ gas and ~700 μl of blood aspirated from the right ventricle using a 21-Gauge needle. The blood was allowed to clot for an hour in Microtainer Serum Separator Tubes (BD Biosciences, San Jose, CA, USA) and then centrifuged at 2,000 r.p.m. for 15 min. The upper (serum) phase was collected and albumin and IgG levels assessed at the Charles River Laboratory using ELISA assay kits (Albumin kit: #E90AL; IgG kit: #E90G, Immunology Consultants Laboratory, Portland, OR, USA).

**Histochemistry.** The cerebral microvasculature was assessed in 50 μm thick coronal sections using an antibody against Glut1 or labelled lectin. Details are supplied in the supplemental methods section. Capillary density and complexity in the various brain regions examined involved quantifying individual and aggregate length of vessels <6 μm in diameter in a 240 μm² area or 0.003 mm³ volume. Three such areas, non-adjacent to one another, in each animal were examined. Hematoxylin/eosin staining experiments were performed on paraffin-embedded 8 μm coronal sections (brain), 4 μm longitudinal sections (liver, heart, kidney and muscle) or 4 μm transverse sections (intestines) and imaged on an Eclipse 80i Nikon microscope (Nikon, Japan).

**BBB integrity.** To examine the integrity of the Glut1 deficient BBB, BSA conjugated to Alexa Fluor-594 (1 mg per 20 g; Life Technologies), FITC-labelled goat anti-human IgG (150 μg/20 g; Jackson Immunoresearch) or TMR-biocytin (1% in PBS; Thermofisher) was injected into the tail veins of mice. Perfused (4% PFA) brains from the mice were harvested 16 h (BSA, IgG) or 30 min (TMR-biocytin) post-injection and 50 μm coronal sections cut on a vibratome. The tissue was then either stained with fluorescently labelled lectin or an anti-Glut1 antibody, and the microvasculature imaged on a confocal Leica TCS SP5 II microscope. To disrupt the BBB, mice were exposed to kainic acid (15 mg kg⁻¹, I.P.) 24 h before injecting them with labelled biocytin, BSA or IgG.

**Cerebral microvasculature.** To examine the cerebral microvasculature, whole brains were extracted following transcardial perfusion with 1 × PBS and 4% PFA. The tissue was post-fixed overnight in 4% PFA and coronal sections (50 μm) cut on a vibratome the following day. The sections were incubated (1 h, RT) in blocking solution (3% BSA, 0.5% Triton X-100 in PBS) before further incubating them at 4 °C overnight (with a rabbit anti-eGFP antibody, 1:1,000, Life Technologies) or for 48 h (with a rabbit anti-Glut1 antibody, 1:1,000, Millipore). They were then washed three times (15 min each) with 1X PBS and an appropriate fluorescently labelled secondary antibody diluted 1:1,000 in blocking solution allowed to bind the primary antibody overnight at 4 °C. Following a second set of washing steps, the slides were mounted (Vectashield, Vector Labs, Burlington, VT, USA) on slides, overlaid with a coverslip and imaged on a Leica TCS SP5 II confocal microscope. Images presented in the manuscript are reconstructions of three dimensional z-stacks. Lectin staining was performed in a similar fashion following treatment of the tissue sections with fluorescein-labelled *Lycopersicon esculentum* lectin (Vector Labs).

***In vivo*, two-photon microscopy.** To image the cerebral microvasculature in live animals, we followed a protocol previously described by McCaslin *et al.*[42] Eight week old mice, administered AAV9-Glut1 at PND3, and relevant controls were anesthetized (1,500 mg kg⁻¹ urethane and 500 mg kg⁻¹ glycopyrrolate, administered intraperitoneally). Next, a craniotomy on the right hemisphere between bregma and lamda was performed, following which fluorescein-conjugated dextran (2,000 kDa, 0.1 ml from 25 mg ml⁻¹) injected into the tail vein to enable visualization of the capillaries. Images were acquired using a home-built two-photon laser scanning microscope[63,64] equipped with a 20 ×, 0.95 NA objective lens (XLUMPLanFl, Olympus). Stacks of angiograms (~510 × 510 μm) were constructed beginning at the cortical surface down to a depth of ~500 μm. Images were acquired every 2 μm in the z axis. Microvascular length was quantified by modifying an image processing pipeline previously described[65]. Image analysis was performed using ImageJ and MATLAB. Briefly, three sub-regions (510 × 510 × 20 μm) at depths of 200, 300 and 400 μm were selected from the stack. The mean image of the sub-region was first pre-processed with a tubeness filter to enhance the features of the vessels. Then, an automatic intensity thresholding was applied to segment the vessels. Capillary diameter was further determined by skeletonization and Euclidean distance map. Only blood vessel segments with diameters of <6 μm were included in the final result.

**Small animal FDG-PET and EEG analysis.** Wild-type ($n=11$), Glut1 DS ($n=12$) and AAV9-Glut1 ($n=4$) animals were fasted for approximately 9–12 h before imaging. Mice were placed in an induction chamber containing ~2% isoflurane/oxygen and then secured to a custom 4-mouse bed for placement of tail vein catheters. Anesthesia (~1% isoflurane/oxygen) and body temperature were maintained during the imaging procedures. The mice underwent dynamic small animal PET imaging from 0 to 60 min after intravenous tail injection of $210-235$ μCi (7.4 − 8.7 MBq) of [¹⁸F]FDG using an INVEON PET/CT system (Siemens Medical Solutions). Computed tomography (CT) images were acquired with the INVEON system on all mice. The PET data was analysed by manually drawing 3D regions of interest over the entire brain using the Inveon Research Workplace software package (Siemens) and uptake expressed as average standardized uptake values. EEG monitoring and analysis were performed using methods detailed by us in a prior study[11].

**Glycolytic flux and mitochondrial respiration studies.** Mutant or WT cells, derived as described in supplemental methods, were directly seeded on the XF 24-plate at a density of $6 \times 10^4$ cells per well in DMEM-F12 (1:1) medium supplemented with 10% fetal bovine serum (FBS) and 2 mM Glutamine. Media was replaced every 3–4 days. After 2 weeks *in vitro* cells were equilibrated with bicarbonate-free DMEM medium (without pyruvate, lactate, glucose, glutamine; Seahorse bioscience) supplemented with 2.5 mM glucose. The OCR and ECAR was measured using a Seahorse XFe24 Extracellular Flux Analyzer (Seahorse Bioscience)[34]. After a baseline measurement, mitochondrial function and glycolytic flux was determined as previously described[66]. Oxygen consumption and glycolytic rates in astrocytes were determined through sequential addition of 6 μM oligomycin, 0.5 mM 2,4-dinitrophenol, and 1 μM antimycin per 1 μM rotenone. After the experiment, protein concentration was determined for each well and OCR and ECAR values were normalized by mg. protein.

**Statistics.** The unpaired 2-tailed Student's *t*-test with Welch's correction or one-way ANOVA followed by Tukey's *post hoc* comparison, where indicated, was used to compare means for statistical differences. To determine if there was a significant alteration in protein concentrations in the CSF of Glut1 DS patients, a one-sample *t*-test was used and a comparison made to the mean value obtained from Biou *et al.*[29] Data in the manuscript are represented as mean ± s.e.m. unless otherwise indicated. $P<0.05$ was considered significant. Statistical analyses were performed with GraphPad Prism v6.0 (GraphPad Software).

**Data availability.** The authors declare that all data supporting the findings of this study are contained within the article and its Supplementary Information Files or available from the authors upon request.

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

# ARTICLE

58. Matsuura, K., Kabuto, H., Makino, H. & Ogawa, N. Pole test is a useful method for evaluating the mouse movement disorder caused by striatal dopamine depletion. *J. Neurosci. Methods* **73,** 45–48 (1997).

59. Gao, G. P. & Sena-Esteves, M. in *Molecular Cloning, Vol 2: A Laboratory Manual.* (eds Green, M. R. & Sambrook, J.) 1209–1313 (Cold Spring Harbor Laboratory Press, New York, 2012).

60. Yardeni, T., Eckhaus, M., Morris, H. D., Huizing, M. & Hoogstraten-Miller, S. Retro-orbital injections in mice. *Lab Anim.* **40,** 155–160 (2011).

61. Glascock, J. J. *et al.* Delivery of therapeutic agents through intracere-broventricular (ICV) and intravenous (IV) injection in mice. *J. Vis. Exp.* **56,** 2968 (2011).

62. Monani, U. R. *et al.* The human centromeric survival motor neuron gene (SMN2) rescues embryonic lethality in Smn(-/-) mice and results in a mouse with spinal muscular atrophy. *Hum. Mol. Genet.* **9,** 333–339 (2000).

63. Galwaduge, P. T., Kim, S. H., Grosberg, L. E. & Hillman, E. M. Simple wavefront correction framework for two-photon microscopy of *in-vivo* brain. *Biomed. Opt. Express* **6,** 2997–3013 (2015).

64. Cebulla, J., Kim, E., Rhie, K., Zhang, J. & Pathak, A. P. Multiscale and multi-modality visualization of angiogenesis in a human breast cancer model. *Angiogenesis* **17,** 695–709 (2014).

65. Pons, R., Collins, A., Rotstein, M., Engelstad, K. & De Vivo, D. C. The spectrum of movement disorders in Glut-1 deficiency. *Mov. Disord.* **25,** 275–281 (2010).

66. Llorente-Folch, I., Rueda, C. B., Pérez-Liébana, I., Satrústegui, J. & Pardo, B. L-lactate-mediated neuroprotection against glutamate-induced excitotoxicity requires ARALAR/AGC1. *J. Neurosci.* **36,** 4443–4456 (2016).

## Acknowledgements

We thank members of the De Vivo, Gao and Monani labs for comments and suggestions, and M. Hirano for the use of the Seahorse Bioanalyzer. We are also very grateful to A. Burghes and H. Zoghbi for critically reading this manuscript. D.C.D, G.G. and U.R.M were supported by a grant from the Will Foundation. Additional support provided by the Hope for Children Research Foundation, Milestones for Children, Glut1 Deficiency Foundation, Crofoot/Walz Family and Joseph Fung Family to D.C.D, P01 AI100263 to G.G., R01 NS063226 and U01 NS094296 to E.M.H, R01 NS029709 to J.L.N, and R01 NS057482, Sanofi-Aventis, Glut1 Deficiency Foundation, University of Pennsylvania Orphan Disease Center and MDA to U.R.M.

## Author contributions

M.T. planned and performed most of the experiments described here. G.G. provided intellectual input and supervised the preparation of all AAV9 constructs. T.A., H.Y., F.L and K.E.S assisted with cell culture, expression studies, uptake assays and mouse beha-vioural assays respectively. H.L. and Q.S. cloned the Glut1 constructs and purified the AAV9 vectors. K.M.E. helped manage the experiments. C.B.R. assayed the expression of the BBB signature genes. C.B.R and M.S.-Q. analysed glycolytic flux in mutant cells. L.J. and J.M. performed the micro-PET studies. R.S. and J.L.N. carried out the EEG analyses. H.Y. and D.T. performed the *in vivo* imaging studies while E.M.H. supervised these experiments. D.C.D. initiated the project, provided intellectual input and helped prepare the manuscript. U.R.M. conceptualized the experiments, directed the project, analysed and interpreted the data, and wrote the manuscript.

## Additional information

**Competing financial interests:** U.R.M., D.C.D., G.G. and K.M.E. have filed a provisional patent application on the use of Glut1 for gene replacement strategies for the human disease.

