## [Peer Review File · Nature Communications]

Reviewers' comments:

Reviewer #1 (Remarks to the Author):

Authors: Maoxue Tang, Guangping Gao, Tomoyuki Awano, Carlos B. Rueda, Kristen M. Engelstad, Hong

Yang, Fanghua Li, Huapeng Li, Qin Su, Kara E. Shetler, Lynne Jones, Jonathan McConathy, Darryl C. De Vivod, and Umrao R. Monani

Title: Brain microvasculature defects and Glut1-deficiency syndrome averted by early repletion of the Glucose Transporter-1 protein

This manuscript investigates the therapeutic effects of temporal treatment of AAV9-constructs in the model mice of Glut1-deficiency syndrome. Glut1 repletion in neonates by using such a vector was found to improve motor performance, reverse low CSF glucose, enable the normal development of the brain microvasculature and prevents microencephaly in Glut1-DS mice. This study demonstrated that insufficiency of Glut1 protein arrests normal neonatal cerebral angiogenesis, whereas timely AAV9-mediated repletion of the protein prevented the diminution of brain microvasculature and thus ameliorates the disease.

There is no cure for Glut1-DS patients and the standard treatment so far is the ketogenic diet, which has been effective in treating seizures, but less effective in cognitive impairment or behavioral issues. Therefore it is promising that AAV9-mediated treatment could restore some of the brain functions and prevent the microencephaly which is one of the main symptoms of Glut1-DS. However, there are some major concerns about this study:

The authors state that an AAV9-mediated viral repletion of Glut1 protein acts mainly through restoration of brain microvasculature. However, the evidences for a restoration of the angiogenesis are not either sufficient or convincing. Firstly, the vasculature parameters from three slices of thalamus per animal are not sufficient for a statement of angiogenesis for the whole brain. What about the capillaries in the other brain regions, such as cortex and hippocampus, which are the brain regions that related closely with motor and cognitive function? Secondly, the perfusion status of the capillaries in a whole brain, as well as in some specific brain regions, should be provided (like a perfusion-weighted MRI study), to confirm a regaining of the capillary function. Thirdly, there is no direct causal link either between Glut1 protein and angiogenesis, or between the regained angiogenesis and the improved motor function. Lastly, since the in vitro study of functional evaluation of Glut1 constructs was done in patient fibroblasts, not in endothelial cells, this data may not support the function of Glut1 expression via AAV9 constructs in endothelial cells of the capillaries in vivo. Generally, although the timely treatment of AAV9-Glut1 is effective, the results from this study do not fully support the statement that angiogenesis is the link between Glut1 repletion and improved brain function.

Minor comments:

1. The methods indicate two routes of virus delivery, systemic and intra-cerebral injection. However, it not been clarified which route has been used for each related experiment.
2. Although one supplementary figure shows no difference between facial vein injection and ICV injection, in terms of blood/CSF glucose and body/brain weight. Due to BBB, usually the local brain injection should yield higher viral expression. Please also provide immunohistochemistry images if there is no difference in Glut1 expression between two injection routes.
3. Seems like no fluorescence being conjugated in the viral construct, so it is very difficult to evaluate the viral expression level in the capillary endothelial cell as well as brain tissue. It would be more convincing to provide the immunostaining of Glut1 for all the quantification of the blood vessels.
4. Seizures occur frequently to the Glut1-DS patients. Since the Glut1-DS model mice also have seizure activity as author states, experiments should be done to indicate how effective the treatment to reduce seizures.

Reviewer #2 (Remarks to the Author):

In this work the authors explore the impact of reduced expression of the Glut-1 protein expression in brain development in genetically targeted mice. The analysis of mice with only one intact Glut-1 gene, and thus a 50% reduction of the protein levels in the microvasculature is a model of the human condition known as Glut1-deficiency syndrome (Glut1-DS). In humans, this condition disrupts normal brain function and cause a severe pediatric defect in brain development. Glut-1-DS is considered as a very rare condition but novel estimates reveal that it might be more prevalent than previously thought. The results in this work show that Glut-1 deficiency in mice causes severe defects in brain development with arrested postnatal angiogenesis leading to a defect brain microvasculature, similar to the human condition. Attempts to restore Glut-1 levels in heterozygous Glut-1 mice using AAV9-mediated gene transduction during early postnatal development resulted in restoration of angiogenesis and prevention of disease development. However, in older mice, the gene transduction had less beneficial effects and in adult mice no improvements were seen. Thus the authors identified a time window in which Glut-1 repletion in early symptomatic animals can restore the cerebral microvasculature and ameliorate the disease.

General; This manuscript describes a set of carefully conducted experiments that addresses a medically important area, namely a treatment paradigm for the devastating consequences of glucose deficiency in the CNS. The results are clearly presented and the conclusions are supported by the experimental data. A number of important control experiments are also included. The manuscript is well written and easy to understand.

Specific comments; I find it surprising that no effect on the integrity of the BBB can be observed since the brain parenchyma obviously suffers from an energy deficient state. The authors use fluorescently labelled macromolecules to reach this conclusion. This is also in contrast to a previous finding in the fish. Have the authors tested tracers of lower molecular weights to more carefully address the BBB integrity and can that explain the differences to previous results?

Secondly, can the authors explain why there are very significant differences in CSF glucose concentrations between the mutant mice, mutant mice + AAV transducer and wt, probably reflecting differential endothelial uptake of blood glucose, and the PET data presented in Fig. 5a. The discrepancy is quite huge and needs to be explained.

Reviewer #3 (Remarks to the Author):

The major findings presented in this manuscript entitled "Brain microvasculature defects and Glut1-deficiency syndrome averted by early repletion of the Glucose Transporter-1 protein" by Tang, M et al are essentially as described in the title. Mice with a haploinsufficiency of the SLC2A1 gene express reduced levels of the Glut1 glucose transporter in the brain and other tissues. This condition results in neuroglycopenia and hypocalcaemia that leads to seizures, delayed neural development, microcephaly and movement disorders. The authors demonstrate that infection of these mice with an aav9 virus expressing Glut1 can prevent/reverse these symptoms if introduced early in the course of neural development i.e. up to postnatal day 14 but was ineffective when introduced in the 8-week adult mouse despite being able to promote increased CNS glucose uptake. They also describe the reduced cerebral microvascular development resulting from the haploinsufficiency, which is not novel as Dr. DiVivo and other colleagues (Winkler et al, 2015) had previously been shown this to be the case. However, in contrast to the previous study, the current study does not find any breakdown in the blood-brain barrier. Given that both studies apparently use the same mice, a more comprehensive discussion of these important

differences should be provided.

General Comments

A major concern this reviewer has with this manuscript is the failure to recognize the significance of the other Glut1 protein (45 KDa) that resides in all glia, which in this mouse model is also reduced by 50%. On Western blots of whole adult brain tissue it represents >65% of the total Glut1 protein (Vannucci, ref 47) that is detected and there is no indication as to whether in the repletion experiments, the Glut1 aav9 virus is repopulating the transporter deficient astrocytes. The blots as presented do not adequately resolve the proteins in the brain samples. The role of astrocytes in both neuronal development and neural homeostasis widely appreciated. Moreover, the growing belief that astrocytes are responsible for providing energy in the form of lactate for neuronal metabolism (see reviews by Magistretti). The loss of 50% of the Glut1 transporters would certainly compromise astrocytic and neuronal growth and would obviously lead to not only energy depletion and seizures but ultimately microcephaly. Astrocytes also play a crucial role in angiogenesis, which is compromised in these mice and is also not considered in the proposed figure 8. The authors proposed that the endothelial cell metabolism is compromised by the transporter deficiency. This would appear to be unlikely as circulating glucose levels are normal and the endothelial cells represent 2% of the brain and require only a very small % of the glucose that is transported into the brain for neuronal and glial metabolism. Compromised astrocytes and pericytes would be more likely to impair angiogenesis. Finally it seems obvious given the time table for neurogenesis in the mouse is essentially complete by 4 weeks that adding back what would be a vital transporter for development 4 or 16 weeks after neurogenesis is completed would be without effect. A more comprehensive description of neural development would put the observations in context-i.e what is the time-table for the development of astrocytes, oligodendrocytes, neurons and endothelial cells.

Specific Comments

Results

In Figure 3 the authors describe the infection of patient fibroblasts with the Glut1 aav9 construct. A couple of questions arise 1) why are there two forms of the transporter and do they really represent a 45KDa and 55 KDa form of the transporter or is this an artifact of gel system. Running a microvessel sample as a control would address this question. 2) Do the control patient fibroblasts increase their transporter level if infected?

There appears to be no distinction as to whether the virus was presented by icv or through the facial vein therefore it would appear that the virus has complete access to the CNS by either route. Can the authors confirm this and demonstrate incorporation of the viral Glut 1 into astrocytes.

P10- not transgenic

P12- 'prevents release into the circulation' needs clarification

Discussion

A discussion of the similarities and differences between the current paper and that of Winkler et al should be expanded.

When considering the human patients it should be noted that under normal circumstances patients are less innately ketogenic than rodents.

Angiogenesis is but one component in the neural dysfunction caused by a lack of Glut1-see general comments. This requires Fig 8 to be revised or scraped

P16. As indicated above the haploinsufficiency is unlike to impede endothelial cell metabolism.

P16 the increase in 45 KDa Glut1 concentration is 5 fold between day14 and 28 whereas the 55 kDa Glut1 is at most 2 fold over the same period (ref 47).

Methods

The resolution of 55 and 45 KDa could be significantly improved to resolve any differential contribution of the viral transporter to the glia and endothelial cell.

Reviewer #1

We were encouraged by the reviewer's prompt acknowledgment of the lack of a truly effective treatment for Glut1-DS and his recognition of the significance of studies to develop an optimal therapy for the disease. Accordingly, we wish to thank him for noting that "it is promising that AAV9-mediated treatment could restore some of the brain functions and prevent the microcephaly which is one of the main symptoms of Glut1-DS." Yet he did raise several concerns most of which centered on a potential link we highlight between Glut1 deficiency and a poorly developed brain microvasculature. We have assiduously attempted to address this concern. Specific points in connection with this criticism and our responses to these follow below:

1. *Comment:* "The vasculature parameters from three slices of thalamus per animal are not sufficient for a statement of angiogenesis for the whole brain. What about the capillaries in the other brain regions, such as cortex and hippocampus, which are the brain regions that are related closely with motor and cognitive function?"

Response: We fully agree with the reviewer's comment and never meant to focus exclusively on the microvasculature of the thalamus. The results from this brain region were emphasized merely because the thalamus, as stated in the original text, was found to be particularly hypometabolic in previous studies conducted by our group. Nevertheless, we ought to have included data from other brain regions and do so now. In particular, we examined the cortex (primary motor cortex and somatosensory cortex) as well as hippocampus (CA1, CA3 and dentate gyrus) noting a similar paucity of capillaries in the vehicle-treated Glut1-DS mutant. Importantly, the complexity of the microvasculature was restored following early repletion of AAV9-Glut1. This not only strengthens the results from the thalamus but also further attests to the link between Glut1 and the capillary network in the overall brain. The results are now included in the section titled, "Brain microvasculature defects in Glut1-DS model mice" – page 4 – and expanded upon in later sections as well as in Fig. 5.

2. *Comment:* "The perfusion status of the capillaries in a whole brain, as well as in some specific brain regions should be provided to confirm a regaining of capillary function."

Response: We address this point by resorting to a well-established *in vivo* imaging technique that exploits 2-photon microscopy and examines the brain microvasculature in *live* animals (McCaslin, A.F.H. et al, 2011, *J Cereb Blood Flow Metab.* **31**:795-806 and references therein). In essence then, the analysis is conducted on *normally* perfused capillaries rather than on fixed tissue. As expected, we found fewer perfused capillaries in vehicle-treated mutants at all depths examined. In contrast, early treatment with AAV9-Glut1 restored the network to a wild-type state. The results now constitute Fig. 5c, e & f, Supplementary Video 1 and a relevant description in the main text (page 11). The new experiments focused on the brain cortex as a matter of necessity; 2-photon imaging allows one to examine cortical capillaries up to depths of ~700µm, but unlike MRI studies one can do so with unparalleled clarity, visualizing the complexity and morphology of individual vessels. We do hope that when combined with our analysis of the various regions of the *fixed* (4% PFA) brain, the data satisfactorily convinces the reviewer of the link between Glut1 and the brain microvasculature.

3. *Comment:* "There is no direct causal link either between Glut1 protein and angiogenesis, or between the regained angiogenesis and the improved motor function."

Response: Although we agree that we failed, in the original manuscript, to conclusively establish a direct link between Glut1 and angiogenesis, we were not altogether dismissive of potential mechanisms connecting one

with the other, speculating in the Discussion section on the possibility that reduced glycolytic flux in endothelial cells perturbs their ability to properly form blood vessels. We expand on this angle in the revision (Fig. 3, Supplementary Fig. 4 and a new paragraph in the text – page 6), demonstrating that Glut1 deficient cells in culture *do indeed* exhibit evidence not only of reduced glycolysis but also of maximal respiration. Perturbations in glycolysis have been clearly shown by the Carmeliet group to inhibit capillary sprouting (Cruys, B. et al, 2015, *Nat. Commun.* **7**: 12240; De Bock, K. et al, 2013, *Cell*, **154**: 651-663). It would not be a stretch to assume, given our new results, that low Glut1 reduces glycolytic flux and that this state conspires with an inability of the endothelial cell to attain maximal respiration, to hinder angiogenesis in Glut1-DS. Reduced glycolysis is clearly sufficient to impede angiogenesis according to De Bock *et al.* However, it may not be the exclusive mechanism linking Glut1 to defects of angiogenesis. In support of this claim, we also draw the reviewer's attention to new data we have generated (Fig. 3) demonstrating a significant decline in levels of the angiogenesis effector Vegfr2 in Glut1-DS endothelial cells (microvasculature). Knockdown of Vegfr2 in endothelial cells has been shown to affect their proliferation (Shalaby, F. et al, 1995, *Nature*, **376**:62-66) and consequently, this molecule could function as a second mediator of poor angiogenesis in Glut1. While we are excited by these findings and believe they address, at least in part, the reviewer's comment, we readily admit that revealing precise mechanisms will require a great deal more work. Teasing out the details of such mechanisms and the exact links between defective angiogenesis and motor abnormalities will involve painstaking examinations of the intricate circuitry of the brain. Such experiments are simply beyond the scope of this article which is mainly meant to highlight the future promise of gene replacement as a means to treat Glut1-DS. We do hope the reviewer agrees.

4. *Comment*: "Since the in vitro study of functional evaluation of Glut1 constructs was done in patient fibroblasts, not in endothelial cells, this data may not support the function of Glut1 expression via AAV9 constructs in endothelial cells of the capillaries in vivo."

Response: We appreciate the reviewer's comment but gently remind him that while the initial functional evaluation was carried out in patient fibroblasts, subsequent analyses involved a detailed, *in vivo* examination of the brain microvasculature – which comprises endothelia – in response to AAV9-Glut1 administration. As emphasized in the paper and in our previous comments, the capillary network and, by inference, the endothelial cells were substantially rescued following such treatment. Additionally, we reiterate the results of our brain glucose uptake studies and CSF glucose measurements. These data constitute major components of our overall findings and, given the established role of brain endothelial Glut1 in the facilitated transport of glucose across the BBB, it is puzzling how each parameter would increase in AAV9-Glut1-treated mutants were it not for a concomitant increase in levels and/or function of the protein in these cells. It is true that glucose uptake, as assessed in fibroblasts, was not conducted using endothelial cells, but we believe that glucose measurements in CSF and brain are an even more physiologically relevant outcome when examining endothelial cell function post-treatment in the context of Glut1-DS. We trust this explanation serves to allay the concern of the reviewer.

5. Minor comments:

- (1) *Comment*: "The methods indicate two routes of virus delivery, systemic and intra-cerebral injection. However, it has not been clarified which route has been used for each related experiment."

Response: We apologize for any confusion and note that the predominant choice of virus administration was systemic. ICV injection was only used to determine if this second mode of delivery also provided therapeutic benefit, which would indicate that such a route of administration also likely targets brain endothelial cells. Still, we address the comment by including a sentence (page 8) stating that, "Unless otherwise noted, subsequent results stem from systemically administered virus."

- (2) *Comment*: "Due to BBB, usually the local brain injection should yield higher viral expression. Please also provide immunohistochemistry images if there is no difference in Glut1 expression between the two injection routes."

Response: We agree that restricted, intrathecal delivery can result in higher levels of virally-derived transgenic constructs. However, this would not be the case – as in our study – if titers were correspondingly lowered vis-à-vis systemic delivery. Accordingly, we did not observe a discernible difference between the two routes of delivery – a result we did not make explicit as we did not think it will alter the core message of our work. Nevertheless, we include below for the benefit of the reviewer, representative images of the microvasculature following either ICV or systemic AAV9-Glut1 delivery. Additionally, we include in the Methods section a note (page 21) indicating the lower titer of the virus used for the ICV injections.

- (3) *Comment*: “Seems like no fluorescence being conjugated in the viral construct, so it is very difficult to evaluate the viral expression in the capillary endothelial cell as well as brain tissue. It would be more convincing to provide the immunostaining of Glut1 for all quantification of the blood vessels.”

Fig. A. Immunohistochemistry of the brain microvasculature of 2-month old mice following AAV9-Glut1 delivery, at PND3, through the cerebral ventricles or bloodstream. Note equivalent expression of Glut1 protein in the two panels, as assessed by fluorescence intensity. Regions denoted by asterisks depict neuropil expressing the virally-derived Glut1 protein.

Response: We agree with the reviewer that inserting a tag into the construct would have enabled us to distinguish between virally-derived and endogenous Glut1 expression. However, such tags often interfere with transgene expression and/or function. Accordingly, we selected not to tag the transgene. Still, based on independent studies (Foust, K.D. et al, 2009, *Nat. Biotech.* **27**: 59-65; Foust, K.D. et al, 2010, *Nat. Biotech.* **28**:271-274) and our own, we establish that AAV9 is capable of targeting brain endothelia *as well as* cells of the brain parenchyma (Supplementary Fig. 6A; also see Fig. A above). While our studies and those of Foust *et al* mostly involved the use of an AAV9-eGFP construct, we have no reason to believe that AAV9 carrying the Glut1 gene behaves any differently. Thus we are confident that both endothelia as well as astrocytes – the two likely sites of action of Glut1 – are targeted in our experiments. We further justify our use of lectin-stained vessels in quantifying the brain microvasculature based on an important result (Supplementary Fig. 1A) that we included in the original text. This result indicated that lectin and Glut1 *invariably* identify the same capillaries. This is true of all brain regions (thalamus, cortex and hippocampus) that we examined and applies equally to mice treated with AAV9-Glut1 (see Fig. B below). Thus we are reasonably certain that the quantification of the capillary network based on our lectin staining experiments is indeed an accurate reflection of the state of the Glut1-expressing, endothelial cell-derived brain microvasculature. We do hope this satisfactorily addresses the reviewer’s concern.

- (4) *Comment*: “Seizures occur frequently in Glut1-DS patients. Since the Glut1-DS model mice have seizure

Fig. B. Immunohistochemistry of the brain microvasculature of 2-month old mice treated with AAV9-Glut1 at PND3. Note that lectin stained structures are in perfect register with Glut1-positive vessels, justifying the use of lectin in quantifying the complexity of the brain capillary network.

activity as author [sic] states, experiments should be done to indicate how effective the treatment [sic] to reduce seizures.”

Response: The reviewer is absolutely correct. Glut1-DS is indeed characterized by epileptic-like seizure activity, and this is true of our model mice. We apologize for omitting this analysis in the original text and now show (Supplementary Fig. 7I, J and accompanying text on page 11) that whereas seizures, as determined by abnormal EEG activity, persist in vehicle-treated animals, they are greatly mitigated in mutants treated with AAV9-Glut1. Owing to the extremely time-consuming nature of these studies, we provide evidence of correction only in animals treated at PND3. We hope the reviewer finds this acceptable.

Reviewer #2

The reviewer was generous with his/her comments noting our “results are clearly presented and [that] the conclusions are supported by the experimental data.” S/he goes on to indicate that “the manuscript is well-written and easy to understand.” Nevertheless, s/he requested two clarifications which we offer below.

1. *Comment:* “I find it surprising that no effect on the integrity of the BBB can be observed since the brain parenchyma obviously suffers from an energy deficient state. The author use fluorescently labelled macromolecules to reach this conclusion. This is also in contrast to a previous finding in the fish. Have the authors tested tracers of lower molecular weights to more carefully address the BBB integrity and can that explain the differences to previous results?”

Response: The reviewer raises a pertinent point and we address his/her queries as follows. First, we did indeed note the difference in BBB integrity between our Glut1-DS model mice and the zebrafish model of the disease developed by Zheng and colleagues. Perhaps the most parsimonious way to explain the discrepancy is to emphasize the difference in the level of Glut1 knockdown in the two model organisms. Whereas the model mice are haploinsufficient and thus expected to express ~50% of the normal protein, the fish, resulting from a Glut1 antisense morpholino knockdown, express only ~10% of the protein. The far greater decline in Glut1 in the fish could well explain the more severe effects on the BBB in this model. We had alluded to this in the original Discussion section and once again draw the reviewer’s attention to this distinction between the two models. Notwithstanding the differences in Glut1 levels in the two models and the varying ramifications they might have on BBB integrity, we resolved, as recommended by the reviewer, to further investigate the barrier in our mice. Accordingly, we substituted labeled IgG and albumin with a much smaller tracer – TMR-biocytin in new experiments. TMR-biocytin (~900 daltons) is at least two orders of magnitude smaller than either albumin or IgG. Consistent with our previous data, we found no evidence of a leaky BBB in Glut1-DS mice. Indeed, there was no difference in staining patterns between wild-type and mutant brain tissue following administration of the tracer whereas copious fluorescence was detected in the parenchyma when the BBB was chemically disrupted with kainic acid. Our overall results, which now include the new data in Supplementary Fig. 3A, argue convincingly for a grossly intact BBB in Glut1-DS model mice. In future, we will extend our analysis by measuring the trans-endothelial resistance of mutant and wild-type brain endothelial cell monolayers as a means of confirming our current findings.

2. *Comment:* “Can the authors explain why there are very significant differences in CSF glucose concentrations between the mutant mice, mutant mice + AAV transducer and wt, probably reflecting differential endothelial uptake of blood glucose, and the PET data presented in Fig. 5a. The discrepancy is quite huge and needs to be explained.”

Response: The reviewer makes an important observation noting an apparent discrepancy between the results of the PET experiments and those involving the measurements of CSF glucose. However, in making the distinction, it is important to bear in mind that the PET images represent a snapshot in time (immediately after administration of the [¹⁸F] FDG) whereas the CSF glucose measurements denote steady state levels. While we believe that this is the most likely explanation for the perceived differences, we also note that in each experiment the AAV9-treated animal displayed evidence of greater glucose uptake than its vehicle-treated counterpart. In this respect, the results obtained from the two experiments are entirely consistent. We conclude by noting that additional experiments to examine the dynamics of glucose uptake by PET scans are planned, but we believe they are a somewhat unrealistic proposition for our present study. We do hope the reviewer concurs.

Reviewer #3

We thank the reviewer for acknowledging the significance of our main result – that the brain microvasculature defects and Glut1 deficiency disease phenotype are indeed prevented/reversed following restitution of the Glut1 protein. Her main concern related to the significance of the 45kDa (astrocytic) Glut1 isoform. This was summarized in the initial comment below. We are grateful to the reviewer for allowing us the opportunity to respond to this as well as her other, secondary criticisms.

1. *Comment:* “A major concern this reviewer has with this manuscript is the failure to recognize the significance of the other Glut1 protein (45 KDa) that resides in all glia, which in this mouse model is also reduced by 50%. On Western blots of whole adult brain tissue it represents >65% of the total Glut1 protein (Vannucci, ref 47) that is detected and there is no indication as to whether in the repletion experiments, the Glut1 aav9 virus is repopulating the transporter deficient astrocytes. The blots as presented do not adequately resolve the proteins in the brain samples.

Response: We apologize for any confusion with regard to the repletion of the astrocytic (45kDa) isoform of the protein, and do not wish to categorically state or imply in any fashion that our AAV9-Glut1 vector fails to restore this particular isoform to our mutant mice. On the contrary, the restoration is expected to be isoform-neutral, raising levels of the 55kDa as well as 45kDa species. In new experiments (Supplementary Fig. 6H), we show that this is indeed the case. Levels of each isoform, which are now adequately resolved on the gel, increase. We complement this new data with an acknowledgment (Discussion section – page 18) of the role of the astrocyte in the microvasculature defects identified. This is further reflected in a modified version of our model – Fig. 8 – in which the 45kDa isoform is now depicted on astrocytes, and references to reduced glycolytic flux in endothelia and/or astrocyte made in the text associated with the figure. We do hope the collective response adequately assuages the reviewer’s concerns.

2. *Comment:* “It seems obvious given the time table for neurogenesis [brain circuitry?] in the mouse is essentially complete by 4 weeks that adding back what would be a vital transporter for development 4 or 16 weeks after neurogenesis is completed would be without effect.”

Response: The reviewer makes an interesting point, and while one might expect the restoration of a protein involved in brain maturation after the neuronal circuitry is normally established to be futile, we cite the intriguing example of another pediatric neurodevelopmental disorder – Rett syndrome. This disease, which is caused by mutations in the MeCP2 protein shares certain features, notably 1) an absence of overt neurodegeneration and 2) microcephaly with Glut1-DS. Considering the profound effects of MeCP2 loss on brain function (selective loss of protein function in the mouse CNS is sufficient to cause a Rett syndrome-like phenotype), one might expect that restoring the protein past the normal window of brain maturation would also be without effect. Surprisingly, this is not the case (Guy, J et al, 2007, *Science*, **315**:1143; Robinson, L et al, 2012, *Brain*, **135**:2699). Repletion of MeCP2 at ~12 weeks, well after appearance of an overt phenotype was nevertheless highly effective in rescuing the disease phenotype. Unfortunately, this does not appear to be the case in Glut1-DS – one reason we thought it best not to harbor any preconceived notions about the relationship between the maturation of the individual cells constituting brain circuits and the therapeutic window of opportunity in such a disease.

3. *Comment:* “In Figure 3 the authors describe the infection of patient fibroblasts with the Glut1 aav9 construct. A couple of questions arise 1) why are there two forms of the transporter and do they really represent a 45KDa and 55 KDa form of the transporter or is this an artifact of gel system. Running a microvessel sample as a control would address this question. 2) Do the control patient fibroblasts increase their transporter level if infected?”

Response: The reviewer makes an astute observation relating to the two fibroblast-derived bands that were recognized by our Glut1 antibody. We intuitively labeled them 45kDa and 55kDa, as we would have done bands derived from brain tissue. However, upon a closer examination and following the reviewer’s suggestion (see

Fig. C. Western blot analysis of Glut1 isoforms in human fibroblasts and rodent brain tissue.

blot in Fig. C), we find that these bands do not correlate to the two widely reported rodent endothelial and astrocytic Glut1 isoforms. We continue to detect two bands in the fibroblast samples, one of which is quite faint (arrowhead). However, neither band migrates at the sizes displayed by the Glut1 isoforms from the rodent brain. We can only conclude that the Glut1 species detected in the human fibroblasts are processed (glycosylated?) in a novel manner. Importantly this does not detract from the main message being communicated in the original Fig. 3a (now Fig. 3e), i.e., that transducing the fibroblasts with our Glut1 constructs increases the expression of the protein. Still, we thank the reviewer for her keen observation and have re-marked the bands in the figure with the simple label “Glut1.” We also note that control cells transfected with our Glut1 constructs do indeed exhibit evidence of increased (~50%, relative to sham-transfected) transporter based on uptake assays. However, since these studies were conducted on CHO cells, we chose not to include them in the manuscript. We hope this is construed by the reviewer as a reasonable decision.

4. *Comment:* “There appears to be no distinction as to whether the virus was presented by icv or through the facial vein therefore it would appear that the virus has complete access to the CNS by either route. Can the authors confirm this and demonstrate incorporation of the viral Glut 1 into astrocytes.

Response: AAV9 when administered either systemically or ICV in neonatal mice is known to efficiently target CNS cells (Foust, K.D. et al, 2009, *Nat. Biotech.* **27**: 59-65; Passini, M.A. et al, 2010, *J. Clin. Invest.* **120**: 1253).

Moreover, our immunohistochemistry experiments using AAV9-eGFP clearly indicate that this serotype *is* capable of infecting astrocytes (Supplementary Fig. 6A). We have no reason to believe that AAV9-Glut1 behaves any differently and do hope that in conjunction with our western blot data showing an increase in the 45kDa band in AAV9-Glut1 treated mice (Supplementary Fig. 6H) the immunohistochemistry result adequately convinces the reviewer of efficient transduction of the astrocytes by AAV9-Glut1.

5. *Comment:* “P10 – not transgenic”

Response: We regret the misleading terminology and have changed this in the text (page 12) to “construct-derived.”

6. *Comment:* “P12- 'prevents release into the circulation' needs clarification.”

Response: We have attempted to clarify the sentence by replacing it with the following statement: “Considering that the measurements were made in fasting animals, one possible explanation of this outcome is that systemic expression of virus predisposes the animals, when fasted, to hypoglycemia— a likely result of high expression of the Glut1 transgene in organs such as liver and muscle where it contributes to elevated glucose uptake thus lowering serum glucose concentrations.”

7. *Comment:* “A discussion of the similarities and differences between the current paper and that of Winkler et al should be expanded.”

Response: We have now added to the Discussion section (page 17), speculating on the possibility that mouse strain-based differences might explain the different phenotypes reported by Winkler *et al* and us. This oft-overlooked factor can have a profound effect on disease phenotypes in model mice. Accordingly, one possibility is that the mixed 129/SvJ-C57Bl/6J background used by Winkler and colleagues, in contrast to the pure 129SvJ strain we utilized, exacerbates BBB phenotypes in the former study. The mixed background in the Winkler *et al* study arose from the necessity of breeding the Glut1-DS model mice to an animal model of Alzheimer’s disease, which was generated on the C57Bl/6J strain background. We make reference to the different strain backgrounds utilized by the two groups in our revised text and do hope it adequately addresses the reviewer’s comment.

8. *Comment:* “Angiogenesis is but one component in the neural dysfunction caused by a lack of Glut1-see general comments. This requires Fig 8 to be revised or scraped [sic].”

Response: We fully appreciate the reviewer’s comment (also see response to Comment #1) and have included new statements as well as a modified Fig. 8 acknowledging the important role that astrocytes might play in triggering the Glut1-DS disease phenotype.

9. *Comment:* “P16 the increase in 45 KDa Glut1 concentration is 5 fold between day14 and 28 whereas the 55 kDa Glut1 is at most 2 fold over the same period (ref 47).”

Response: We regret the error and have now altered the sentence (page 18) to reflect the precise findings in the article by Vannucci and Vannucci.

10. *Comment:* “The resolution of 55 and 45 KDa could be significantly improved to resolve any differential contribution of the viral transporter to the glia and endothelial cell.”

Response: We have now added the results of a second western blot experiment (Supplementary Fig. 6H) in which the bands are more clearly resolved and demonstrate an increase in *both* the 45kDa and 55kDa band in the AAV9-Glut1-treated mice.

We do sincerely hope that the explanations provided in this letter, the results of our new experiments and the collective modifications to the text satisfactorily address the reviewers' concerns.

Reviewers' comments:

Reviewer #1 (Remarks to the Author):

The additional evidences as well as the thorough explanation that provided by the authors in this re-submission has addressed the majority of my concerns. I would suggest to accept this manuscript.

Reviewer #2 (Remarks to the Author):

In the revised version of the manuscript the authors have provide novel data and responded well to the questions raised by me and the other reviewers. The minor typo on line 439 should be corrected and in Fig.3e the labeling of the lanes should be properly aligned. I have no further comments.

Reviewer #3 (Remarks to the Author):

This manuscript represents a resubmission of a manuscript entitled "Brain microvasculature defects and Glut1-deficiency syndrome averted by early repletion of the Glucose Transporter-1 protein"

General comments

The authors have addressed many of the reviewers' comments but others remain unanswered. The most glaring is the failure of the authors to grasp the role of the endothelial cell to deliver glucose to the brain. The endothelial cells represent 2% of the cells in the brain, the other 98% is made up of glia and neurons which are inherently more active and the glucose that they require passes through the endothelial cells. Even in the mutants where the level of transport is reduced by 50% and CSF glucose is lowered and neurons and glia are effectively energy depleted they are still responsible transporting 25 X more glucose than they require to maintain endothelial cell metabolism. To accomplish this feat of energy supply, they are equipped with approximately 20-30 times the concentration of GLUT1 glucose transporters than the astrocytes. Thus the notion that endothelial cells are energy deficient as the underlying cause for the failure to undergo angiogenesis is not tenable despite the vigorous rebuttal to reviewer 1. The disparity in the levels of transporter could be immediately demonstrated if the western blot used for demonstrating the VEGFRs was re-probed for GLUT1. In part of that rebuttal the authors remark that both astrocytes and fibroblast show reduced metabolic activity. It is precisely this reduced metabolic capacity in astrocytes and pericytes due to their lower [GLUT1] and lower ambient interstitial glucose that is a more likely the cause of impaired angiogenesis. The authors report a reduced Vegf2r in the endothelial cells but should note that astrocytes are an important source of VEGF and are responsible for providing vascular scaffolding and endothelial cell survival. Note Figure 8 should be modified to indicate the disparity in GLUT1 between endothelial cells and astrocytes.

This reviewer also found the response to reviewer 2 comment' regarding the disparity between the reduction in transport and PET unsatisfactory. What should be noted is that PET is a measure of glucose metabolism in the astrocytes and neurons not transport and what is observed is the overall level of metabolism is reduced only moderately because the animals were anesthetized and consequently the energy requirement could still be fulfilled by the lower ambient glucose.

Specific Comments

Abstract

The cognitive dysfunction is undoubtedly due insufficient glucose to enable neuronal function, which the authors have published on extensively.

L.278 this should point out PET changes is in anesthetized animals and CSF glucose reflect free moving animals.

L445-458 needs to be reconsidered and other references to reduced glycolysis that is very unlikely

based on discussion above and as pointed by reviewer 1 was not measured.

The western blots presented in Figure 3 are really misleading as there is no distinction made between 45 KDa astrocyte glucose transporter and the much lighter 55 KDa band which is the endothelial cell GLUT1 and the major focus of the study. Putting the resolution of the two in the supplement and not referring at all to the significance of the two bands is clearly inappropriate. As suggested above re-probe the Vegfr blot to illustrate the relative changes in both in the vessels and vessel free membranes along with a whole brain fraction where they are resolved and the relative concentrations can be appreciated.

Reviewer #1

1. *Comment:* “The additional evidence as well as the thorough explanation that [was] provided by the authors in this re-submission has addressed the majority of my concerns. I would suggest accepting this manuscript.”
Response: We are pleased that we were able to satisfactorily address the reviewer’s concerns and delighted that s/he now finds the article suitable for publication in the journal. We are grateful to the reviewer for his/her valuable input.

Reviewer #2

1. *Comment:* “In the revised version of the manuscript the authors have provided novel data and responded well to the questions raised by me and the other reviewers. The minor typo on line 439 should be corrected and in Fig. 3e the labeling of the lanes should be properly aligned. I have no further comments.”
Response: We are grateful for having had the opportunity to respond to the Reviewer and glad that s/he too now finds our article acceptable. As suggested, the labeling in Fig. 3e has been rectified.

Reviewer #3

We appreciate the reviewer’s continued concerns and attempt, to the best of our ability, to address them in this second revision.

1. *Comment:* The reviewer objects to speculations in the Discussion section that the angiogenesis defects we observe might stem from defects in endothelial cells, suggesting instead that reduced metabolic activity of astrocytes is a “more likely” trigger.
Response: We appreciate the reviewer’s viewpoint and begin by reiterating statements in the Discussion section of the manuscript explicitly acknowledging a possible role for astrocytes and/or pericytes in precipitating the microvasculature defects that were observed (page 18). These, we noted, could have been triggered in the endothelial cells in a non-cell autonomous manner by perturbations originating in astrocytes or pericytes. However, given our current, somewhat limited understanding of angiogenesis defects in Glut1 deficiency, the exact contribution of various cell-types (endothelial cells, pericytes, astrocytes) to the abnormalities and the precise mechanisms operating within the cell-types to arrest capillary development can only be speculated on. Indeed, as acknowledged by the reviewer, the role of astrocytes in triggering the arrest of angiogenesis is at best “more likely” rather than absolutely certain. To unequivocally state that one cell-type or another is at the root of the microvasculature defects, notwithstanding Glut1 expression differences between them, will require considerable additional work that is simply beyond the scope of the current article which is primarily a proof-of-concept study of the mitigating effects of restoring Glut1 in Glut1 DS. We nevertheless make an earnest attempt to address the reviewer’s comment and do so in two ways. First, we re-modify Fig. 8 by adding a note to the legend acknowledging the much greater expression of Glut1 in endothelial cells relative to astrocytes. The endothelial cells *are* shown in the cartoon to have more transporters than the astrocytes, but given the obvious limitations of depicting them using this medium, the relative levels of Glut1 in the two cells could be misconstrued. We hope the note in the legend clarifies and adequately addresses this shortcoming. Secondly, as suggested, we now include blots of Glut1 in the capillaries and brain parenchyma fractions illustrating the relative levels of the protein in endothelial cells vs. neuropil. This constitutes part of Fig. 3b – the blot in which we examined Vegfr2 levels - and depicts a low as well as high exposure time. The new data confirms earlier findings that endothelial cells express much higher levels of Glut1 than does nervous tissue. However, it does not allow one to unequivocally infer if the protein, particularly in mutant endothelial cells, is properly localized to the membrane and thus of functional consequence. Mis-localization of Glut1 from the luminal to cytoplasmic or abluminal membrane domains would obviously affect entry of glucose into the cell.

2. *Comment:* “The authors report a reduced Vegf2r in the endothelial cells but should note that astrocytes are an important source of VEGF and are responsible for providing vascular scaffolding and endothelial cell survival.”
Response: We thank the reviewer for this important information and examined VEGF expression levels in the neuropil of Glut1 DS mutants and wild-type controls at two different time points – PND14 and PND60. At neither point in time did levels of VEGF change significantly in the mutants, suggesting that perturbations in concentrations of this ligand are unlikely to have a major/direct effect on angiogenesis arrest in Glut1 DS mutants. We include below a graph representing this new data.

Fig. 1 - VEGF levels in capillary depleted brain fraction of Glut1 DS mutants and wild-type control animals.
 Note: n ≥ 4 mice of each genotype; t test.

3. *Comment:* “The cognitive dysfunction is undoubtedly due insufficient glucose to enable neuronal function, which the authors have published on extensively.”
Response: We regret the confusion with regard to this statement. The sentence refers specifically to the dearth of information associated with possible (now conclusive) brain pathology underlying Glut1 DS and how such pathology might trigger cognitive dysfunction.
4. *Comment:* “L278 this should point out PET changes is in anesthetized animals and CSF glucose reflect free moving animals.”
Response: As recommended we now make reference to the anesthetized state of the animals undergoing PET scans (page 10). We also clearly state in the revised text that CSF glucose measurements reflect levels in freely moving animals (page 21)
5. *Comment:* “L445-458 needs to be reconsidered and other references to reduced glycolysis that is very unlikely based on discussion above and as pointed by reviewer 1 was not measured.”
Response: As alluded to in our response to Comment #1, we believe, given our current understanding of the link between Glut1 and angiogenesis, that it is impossible to unequivocally state which cell type(s) is/are responsible for the defects. Accordingly, we respectfully maintain the possibility that the microvasculature defects may arise in astrocytes and/or endothelial cells.
6. *Comment:* The reviewer objects to the placement of a blot depicting the 45kDa and 55kDa Glut1 bands in the Supplemental Information.
Response: As suggested, we have moved the above referenced blot from the Supplemental Information to the main text where it now constitutes Fig. 4d. Moreover, we draw the reviewer’s attention to a statement in the manuscript indicating an increase in *both* the 45kDa and 55kDa bands following systemic Glut1 repletion (page 9).
7. *Comment:* The reviewer suggests re-probing the Vegfr2 blot in “Figure 3” for Glut1.
Response: As suggested, and as referred to in our response to Comment #1, we now show two exposures of the Vegfr2 blot (Fig. 4b) re-probed for Glut1. Corresponding reference to the new data is made in the relevant figure legend.

We thank Reviewer 3 for his/her insightful comments and do hope the explanations above and the revisions to the text satisfactorily address all remaining concerns about our article.

REVIEWERS' COMMENTS:

Reviewer #3 (Remarks to the Author):

I would like to state at the outset that I would like to see this paper published as I feel it make an important contribution to the field and the inconsistencies that are being pointed do not take away from the overall strength of the paper but do confuse the potential interpretations. However, this represents the third review of this manuscript and as such I am going to confine myself to the salient points of my previous critics that have still not been addressed.

The authors appear to have a reluctance to acknowledge that there are two forms of the GLUT1 glucose transporter in the CNS. The endothelial cells express the 55 KDa form and the astrocytes, microglia, and pericytes express the 45Kda. In terms of cellular concentration the endothelial cells express a greater level than than the other cells (10-30fold) as illustrated in amended Figure 3 but in terms of total CNS GLUT1, the 45 Kda isoform is significantly more abundant than the 55 KDa isoform as illustrated in Figure 4. The same applies for the mRNA determination is made its origin cannot be distinguished between that the respective cells without fractionation. Similarly, when repletion occurs both the 45 and 55 KDa isoforms are repleted. The problem then arises with the western blots and mRNA in Figure 4 in that what is illustrated and quantified in the upper blot is primarily the astrocytic 45 KDa band which is not stated in the manuscript and consequently misleading. The mRNA determination is clearly the total GLUT1 mRNA of which most would be derived from the cells expressing the 45 KDa isoform. All of this confusion could be alleviated by showing the wild type whole brain expression of both 55 and 45 KDa on the bottom blot with the same resolution and is not apparent in the top blot and consequently misleading. Indeed it would then be possible to determine if the virus differentially infected the endothelial cells, astrocytes, or perhaps neurons.

Having acknowledged the number of transporters in the endothelial cells is significantly higher than astrocytes (see above and p.40) and that in the mutant while CSF levels are glucose is lower than wild, the endothelial transporters in the mutant are still working albeit at half capacity. This reviewer has repeatedly suggested that the endothelial cells cannot be glucose deprived as the endothelial cells would use but a tiny fraction of the glucose that passes through them which has not been addressed. Using astrocytes as a model for endothelial cells having clearly demonstrated that they have far fewer transporters as a model is clearly flawed. Showing that the astrocytes are compromised is important as this would limit angiogenesis and reduce potential astrocyte release of lactate but cannot be equated to metabolism in endothelial cells. Equally a pfk mutation would impair ATP production - reducing level of transporter by a factor of two would have no effect on endothelial cell metabolism as the cellular levels of glucose would still saturate hexokinase and without any further impairment would metabolize normally.

The authors suggest that the reduced expression of vegfr in the mutant mice may underlie the reduced angiogenesis but it would be important to demonstrate that its expression is restored in the mice with the repleted GLUT1

Reviewer #3

We once again thank the reviewer for her time and effort. Her comments and our detailed responses appear below.

- Comment:** The reviewer claims that there is “a reluctance to acknowledge that there are two forms of the GLUT1 glucose transporter in the CNS” and that it is unclear which form is ultimately being restored.

Response: We regret the reviewer’s perception, noting that the purpose of the work was not to focus on any particular Glut1 isoform and that the repletion strategy we chose was never meant to enhance levels of one isoform at the expense of the other. This was made abundantly evident by 1) Stating (see Results section) that regulatory elements of the ubiquitously expressed chicken β -actin gene were used to drive the expression of the Glut1 construct in our AAV9 vector and 2) That the systemically administered AAV9 vector targets endothelial cells (ECs) *and* astrocytes – the sources of the 55kDa and 45kDa isoforms respectively (Suppl. Fig. 6A). Still, to address the reviewer’s concern, we now indicate in the legend to Fig. 4 that quantification of Glut1 in the various animal cohorts takes into account both astrocyte as well as endothelial cell-derived protein. Unfortunately, the related western blots will not, as suggested by the reviewer, allow one to gauge relative transduction in individual subsets (astrocytes and neurons) of cells of the brain parenchyma. We hope the new clarification here and in the text addresses the reviewer’s comment.
- Comment:** The reviewer claims that given the concentrations of Glut1 in ECs, haploinsufficiency of the protein is simply incapable of reducing glucose concentrations within them to affect their metabolism. Additionally, she comments that “using astrocytes as a model for endothelial cells having clearly demonstrated that they have far fewer transporters as a model is clearly flawed.”

Response: We appreciate the reviewer’s remarks and respond by 1) Reiterating a possibility we noted in our previous rebuttal and 2) Highlighting established facts about EC sub-types. First, while it is true that ECs have high concentrations of Glut1, this does *not* automatically mean that they are all appropriately localized, particularly in disease conditions, in the luminal and abluminal membranes to effect glucose transport. It is well-established that even in WT endothelial cells, Glut1 protein is not all inserted into the membrane, but instead distributed in a ratio of 12% (luminal): 40% (cytoplasmic): 48% (abluminal) in these three cellular domains (Farrell and Pardridge, 1991, *PNAS*, **88**:5779) and, furthermore, that such distribution is subject to all manner of physiological perturbations including insulin signaling and cell stress (Widnell, C.C. *et al*, 1990, *FASEB J.* **4**:1634; Palmada, M., *et al*, 2006, *Diabetes*, **55**:421; Ramlal, T. *et al*, 1988, *Biochem Biophys Res Commun*, **157**:1329). Such perturbations could very well operate in Glut1 DS, further depleting *membrane-bound* Glut1 beyond the expected 50% that results from the haploinsufficiency. The combined effects would allow for the very real possibility that haploinsufficiency disproportionately affects glycolytic flux not just in brain parenchyma cells but also in ECs. A second factor that *must* be considered when describing metabolism of brain ECs is to note that these cells are not all homogeneous, particularly during microvasculature expansion. Rather, they consist of at least three sub-types – tip cells, stalk cells and phalanx cells (Potente, M. *et al*, 2011, *Cell*, **146**:873). Phalanx cells are mature, quiescent and fully perfused with blood, serving primarily as conduits to transport blood nutrients to the brain parenchyma and clear away by-products of neuronal/astrocyte metabolism. Admittedly, these are cells unlikely to suffer physiologically meaningful changes in glycolytic flux to alter their metabolism in Glut1 DS, *assuming* that the remaining transporters continue to be properly localized to the luminal and abluminal membranes. In contrast, tip cells which develop filopodia and lamellipodia to form new vessel sprouts and an expanded capillary network into avascular tissue, operate in a quite different microenvironment and, accordingly, could be expected to respond differently to low Glut1. These cells exist in a non-perfused state and are thus exposed to levels of glucose that are close to an order of magnitude lower than those seen by phalanx cells (Abi-Saab, W.M. *et al*, 2002, *J. Cereb Blood Flow Metab*, **22**:271). If the model of Barros *et al* (Barros, L.F. *et al*, 2007, *Glia*, **55**:1222) wherein a 50% loss in Glut1 results in a 90% drop in brain glucose is valid, then in Glut1 DS the endothelial tip cells would be exposed to even lower levels of glucose. It would then not be surprising for glycolytic flux in such tip cells, dually burdened with half as much Glut1 *and* profoundly low

environmental glucose, to be directly affected thus disrupting angiogenesis. This was implicit in our discussion of the results of our findings and it was expected that the expert reviewer would have considered it. To avoid doing so is to ignore established research findings pertinent to the speculation in our discussion. Still, in deference to the reviewer's comments, we have extensively modified the discussion making certain not to refer to glucose-deprived ECs while emphasizing the role of the tip cells in microvasculature formation. Finally, we note that one is *not* equating metabolism in astrocytes to metabolism in ECs. Rather, we extrapolate *relative* metabolism in mutant versus WT astrocytes to *relative* metabolism in mutant versus WT ECs. If one assumes that the percent decrease in Glut1 is equivalent in the two types of cells in Glut1 DS, then we believe that the extrapolation is justified. We hope our aggregate response allays, once and for all, this remaining reviewer's concerns.

3. *Comment:* "The authors suggest that the reduced expression of vegfr2 in the mutant mice may underlie the reduced angiogenesis but it would be important to demonstrate that it [sic] expression is restored in the mice with the repleted [sic] GLUT1."

Response: The Vegfr2 data was not an original concern of the reviewer's and was included in response to a comment from an independent reviewer (Reviewer #1). That reviewer appeared fully satisfied with our response. We hope the matter rests there, particularly (and as recognized by the two other reviewers who now find our manuscript eminently suitable for publication) as it is decidedly peripheral to the crux of the current article which is to highlight the feasibility and promise of gene therapy in Glut1 DS. We hope the reviewer agrees and is further assuaged by our fullest intent to address the link between Vegfr2 and Glut1 in future studies.

We reiterate our sincere gratitude to the reviewer and trust that the explanations provided above and in the main text fully address all her comments.